# Constraints on green renovation of building based on the theory of whole lifecycle and green development

Qiaohui Tong[1], Wei Wei[1]*, Yekai Le[2]

**1** School of Urban Design, Wuhan University, Wuhan, China, **2** WISDRI City Construction Engineering & Research Incorporation Ltd., Wuhan, China

* smiles_weiwei@163.com

## Abstract

Green renovation of building (GRB) is usually full of challenges and limited by multiple factors, including environment, technology, materials, and region. It is necessary to lower the impact on residents' lives and reasonably address construction waste during the renovation process. In this study, a system dynamics model was constructed according to the whole lifecycle theory and green development theory to evaluate the comprehensive benefits of GRB. This model was adopted to analyze the constraints during the renovation process. In addition, the key indicators, were simulated, including pollution, greenness, and resident satisfaction. The system dynamics model simulation results indicated that during the decision-making and project phases, the simulated greenness value of GRB reached 1.623, with a variation value of 1.515. This finding substantiated the substantial influence of this phase on greenness transformation. During the engineering warranty and post-evaluation phase, the greenness change value was the highest of 6.173. Therefore, long-term maintenance and continuous improvement are crucial for greenness. Meanwhile, the proportion of green energy usage gradually increased during the GRB process, while other energy consumption and pollution indicators decreased. This demonstrated the potential of GRB in promoting sustainable development. The study has validated the efficacy of the simulation and analysis of GRB projects through the system dynamics method while providing theoretical foundations for the planning and implementation of GRB projects. By comprehensively considering technological, economic, policy, and social factors, system dynamics model can assist decision-makers in optimizing GRB strategies, achieving a dual improvement in environmental and economic benefits.

## Introduction

Green development is an economic growth and social development model that aims to achieve high efficiency, harmony, and sustainability. It has four important

**Data availability statement:** Data is within the manuscript itself.

**Funding:** The author(s) received no specific funding for this work.

**Competing interests:** The authors have declared that no competing interests exist.

characteristics, including greenness, symbiosis, refinement, and long-term development. Using green materials and energy-saving methods to achieve green renovation of building (GRB) is also a type of green development [1,2]. GRB is limited by the built environment, technology, materials and regions, and is fraught with challenges. Many old buildings have outdated structures and materials, and the backward economic and development conditions are a serious challenge for the design and construction of programs [3,4]. Moreover, the impact of dust and noise on the surrounding residents should be avoided in the renovation process [5]. A lot of construction waste will be generated during the ordinary renovation process and pollute the surrounding soil, water, and air. The research background of the application of whole lifecycle theory (WLT) and system dynamic model (SDM) in green transformation mainly focuses on two points. One is how to minimize environmental impact and resource consumption throughout the entire lifecycle of a building, and the other is how to optimize building performance through dynamic simulation. The application of these theoretical models is of great significance for promoting the sustainable development of the construction industry. In the context of urban renewal and renovation of old residential areas, the application of WLT and SDM is particularly important. In the context of the strategic goals of "carbon peak and carbon neutrality" and the era of green development, the green and low-carbon transformation of old urban communities is crucial for promoting urban renewal, enhancing urban cultural image, and protecting people's living environment. It is a necessary means to achieve sustainable development of existing buildings.

GRB plays a significant role in promoting the development of green concepts. Currently, many researchers have conducted extensive studies and analyses on GRB. Kwame E S and Julian J S used a scope literature review and sequential mixed research methods to explore the parameters that influenced South Africa's decision to adopt green buildings. They determined how incentive mechanisms promote the adoption of green buildings. The relationship between incentive payments, rebates, and discounts for public utility energy efficiency projects and corporate performance was proved [6]. ShengYuan W proposed the application of evolutionary game theory to determine the interactive mechanism of multi-agent behavior in green renovation of commercial buildings in China. The aim was to address the issues of insufficient interaction and low-renovation rates among multiple stakeholders (government, developers, and users). The results showed that government financial support could effectively stimulate developers to actively engage in renovation behavior. Enhancing the whole lifecycle awareness of developers was a key driving force in supporting green transformation and socially sustainable development [7]. John D explored the factors that motivated homeowners, investors, government agencies, and others to engage in GRB by designing a survey questionnaire to address unclear driving forces and slow renovation actions. The questionnaire was distributed to energy consultants, architects, cost engineers, facility managers, and engineers with rich professional experience. The results showed that respondents had a high degree of consensus on the driving factors of green transformation. These included incentive and support systems, violation penalties, high energy bills, energy-saving policies

and regulations, and environmental awareness [8]. Le L addressed the problem of high resource and energy consumption caused by the lack of effective design and construction management methods. From the role of building information model (BIM) technology in optimizing green building performance, the green building performance optimization solution was analyzed in detail. The results showed that this method improved the energy efficiency of green buildings and made the energy cycle in the building more sustainable [9]. The performance and design of building control instrumentation processes were influenced by their evaluation information management and BIM-based model checking. In response to this issue, Atfi Zi proposed a theoretical framework for analyzing the concept of BIM tools and technologies based on GB maintenance environment, thereby demonstrating the feasibility of the model [10]. Vareilles É proposed the method of using Constrained Satisfaction Problem (CSP) to address the issue of high-performance renovation of apartment buildings. The results showed that low-energy consumption performance was achieved through the use of new thermal cladding, while supporting decisions on renovation definition, material list, and on-site assembly process [11]. Lin Y proposed to address the issue of energy-saving renovation of old buildings in the process of urbanization, taking the old brick campus of a university in Chengdu as a case study. Through on-site research and analysis of building functions, structures, and appearance issues, detailed plans for functional upgrades, structural reinforcement, and facade renovation were developed, while integrating solar photovoltaic technology. The results showed that after the renovation, the expected annual solar energy collection was 164,066 kWh, and the self-use electricity of photovoltaic roofs accounted for 42–76% of the total energy consumption, saving about 60% of energy. The economic payback period was about 1.9 years, which had higher economic feasibility [12].

Stopps H proposed a system dynamics (SD)-based approach to address the systemic obstacles in improving the renovation process of high-rise residential buildings using causal circuit diagrams to represent important variables and their causal relationships. Policies that need improvement included mandatory indoor air quality standards, incentivizing comprehensive design, enhancing practitioner education and training, implementing feedback mechanisms, and simplifying material and design certification [13]. Ma W proposed an integrated SD and lifecycle assessment (LCA) method to address the complexity of carbon emission assessment in building renovation waste management. It was found that by establishing five causal cycle models, it was possible to more comprehensively evaluate the carbon emissions of construction renovation waste at different stages. This can help decision-makers gain a clearer understanding of current waste management issues and strategically reduce carbon emissions throughout the entire lifecycle of construction waste [14]. Ouyang T proposed a two-level SDM approach to address the housing stock renovation strategy of French social housing companies in achieving their zero carbon emission goals in 2050. This model provides general renovation directions for housing stock and specific renovation suggestions for each room type. By integrating current regulations and commonly used indicators, the model could more realistically represent actual cases. By interacting with the model and running multiple scenarios, it provided decision-makers with a clear perspective. This helped them better understand and address current issues related to building renovation waste management, thereby strategically reducing carbon emissions throughout the lifecycle of building renovation waste [15]. Nita performed qualitative data analysis and content analysis to address system components and decision-making factors in Indonesia's energy efficiency improvement projects. A comprehensive model was established using SDM and LCA methods to evaluate the carbon emissions of building renovation waste. Through case studies, a causal circuit diagram was established to demonstrate the decision-making factors influencing the adoption of energy efficiency projects and identify the system components, including stakeholders, policies, structures, activities, and resources. The research results indicated that organizational commitment, financial resources, and technology drive were the three main factors affecting the adoption of energy efficiency projects [16].

These studies provide multidimensional perspectives for understanding the complexity of GRB and provide theoretical foundations and empirical data for current research. The research by Kwame E S and Julian J S emphasized the role of incentive mechanisms in promoting the adoption of green buildings in South Africa. This was crucial for understanding how policies affect the green transformation of the construction industry. ShengYuan W used evolutionary game theory to

analyze the multi-party interaction in green renovation of commercial buildings in China. This study revealed the incentive effect of government financial support on developers' participation in renovation behavior, which had guiding significance for formulating effective policy intervention measures. John D explored the driving factors of GRB through a questionnaire survey, which helped identify and strengthen the key factors that promote GRB. Le L et al. explored how to optimize the performance of green buildings from the perspective of BIM technology. This was of great significance for improving building energy efficiency and achieving sustainable energy cycling. Vareilles É and Lin Y respectively demonstrated the potential of achieving low-energy buildings through new technologies and methods from the perspectives of high-performance renovation and energy-saving renovation of old buildings in the process of urbanization. These studies provided specific technical solutions and demonstrated the actual effects of these solutions through case analysis. Finally, the study of SDM provides a comprehensive management framework for GRB, and the effectiveness and environmental benefits of the model have been verified through simulation experiments. The findings of these studies are interrelated and together form a knowledge system in GRB, providing rich background and references for current research [17]. Given this, this study was performed based on the WLT. Comprehensive auditing and monitoring are carried out from project planning to completion and settlement to ensure the comprehensiveness and real-time nature of green renovation. Subsequently, the dynamic changes in the GRB process, as well as the interactions and feedback mechanisms among different factors, were intuitively simulated and evaluated by constructing SDM. It is expected that this method can effectively evaluate limiting factors in GRB, thereby providing a theoretical reference for the green building industry.

However, relevant research lacks a whole lifecycle consideration of building green transformation. Comprehensive auditing and monitoring have not been conducted to ensure the comprehensiveness and real-time nature of green transformation. Without a SDM to intuitively simulate and evaluate the dynamic changes, interactions, and feedback mechanisms among factors in the process of green renovation, it is difficult to effectively evaluate the limiting factors in green renovation. Moreover, comprehensive and targeted theoretical references cannot be provided for the green building industry in this study. The study verified the effectiveness and environmental benefits of the model through simulation experiments. Furthermore, the study examines the alterations in construction waste across the various phases of renovation. Additionally, it presents the findings of simulations about the specific resource energy consumption, pollution, and environmental impact of GRB projects, thereby providing a scientific foundation and strategic guidance for decision-making and the implementation of GRB projects.

## Methodology

In current study, the influencing factors of GRB were first explored, and a SDM was constructed for more in-depth research based on the causes of each factor and their interrelationships. The influencing factors of GRB were discussed. Afterwards, the kinetic model was constructed based on the green development theory (GDT).

### WLT-based management system

**Green construction management organization system based on the whole life cycle theory.** GRB system is a comprehensive building renovation concept and aims to improve the energy efficiency, environmental performance, and sustainability of existing buildings through a series of technical measures and innovative means [18]. The development of GRB systems comes from the increasing global focus on energy conservation, emission reduction, and sustainable development, especially in the construction sector, which is a major energy consumer. WLT has a central place in the GRB system, stressing the need of auditing and monitoring at every phase, from project design and development to construction and settlement. The thoroughness and real-time nature of GRB may be guaranteed by developing a system based on the whole lifespan. Through the conceptual analysis and transformation history of GRB, various roles and responsibilities involved in the GRB project are sorted out, thereby constructing the management organization system. The GRB management organization system based on WLT is proposed in Fig 1 [19].

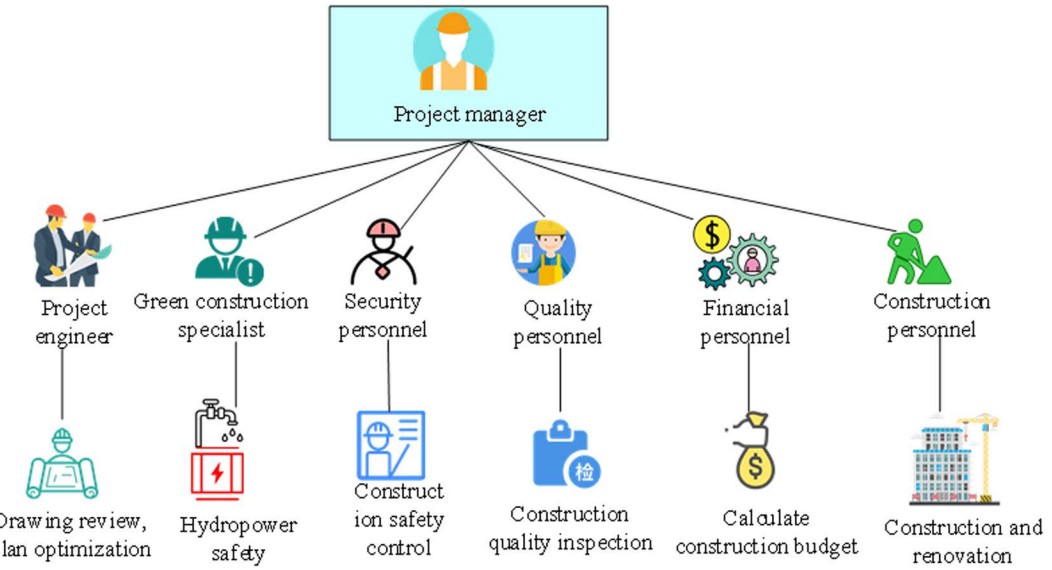

**Fig 1. Management organization system for GRB based on WLT.**

As shown in Fig 1, the project manager serves as the core of GRB and is fully responsible for the GRB project, arranging and controlling the processes and details of various tasks in WLT. The project engineer is mainly responsible for the professional design and optimization of the construction plan. Green construction specialists communicate and work with the project engineer and other personnel. Their work is mainly to participate in the design and evaluation of the project engineer's design program to ensure that the construction design program is green and energy-efficient. In addition, green construction specialists need to collect and analyze GRB data and are responsible for promoting GRB. Safety personnel are accountable for construction safety, eliminating safety hazards, and managing and controlling the construction site. Quality personnel are responsible for on-site construction quality testing. Financial personnel are responsible for budget-related calculations and accounting. Construction personnel are responsible for carrying out specific renovation methods and technologies. This WLT-based system is applicable to all types of building renovation projects. Whether it is commercial or residential, as long as the design of the project conforms to environmental awareness and sustainable development principles, it will be adopted.

**Analysis of constraints on green building renovation.** The whole renovation process is usually accompanied by energy consumption and pollution (Fig 2) [20].

The implementation and renovation of construction activities involve rigorous planning and processes (Fig 2a). The standard is proposed based on the in-depth observation and analysis of the construction activities of the building green transformation, and summarizes the strict planning and process in the implementation and transformation process. This includes the professional audit, on-site inspection, technical requirements, management system formulation, and supervision and management of the link induction. Firstly, it is necessary to develop professional personnel or authoritative institutions review and verify the building renovation plan to ensure the rationality of the design plan. Professional personnel conduct on-site inspections to ensure that its internal environment and physical conditions are suitable for renovation, and solicit opinions from surrounding residents [21]. Renovation mainly focuses on technical reliability and feasibility. GRB requires as many raw materials and green materials as possible and depends on several specialized technical requirements. The flowchart in Fig 2a represents various stages of green renovation of existing buildings, reflecting the entire process of GRBs. It is consistent with the construction of a management system based on the tWLT in research. Auditing

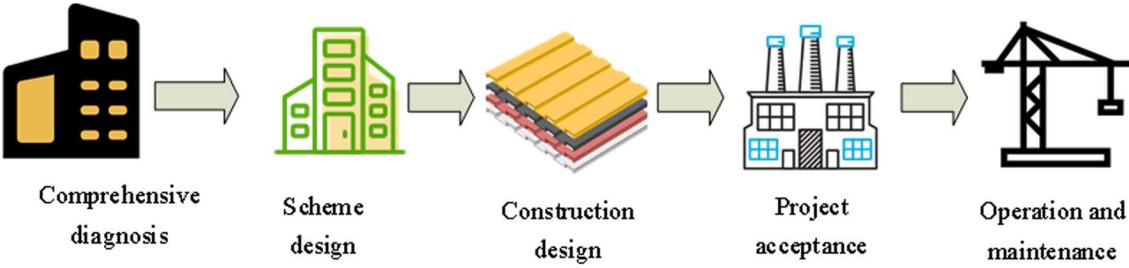

(a) Flow chart of green transformation of existing buildings

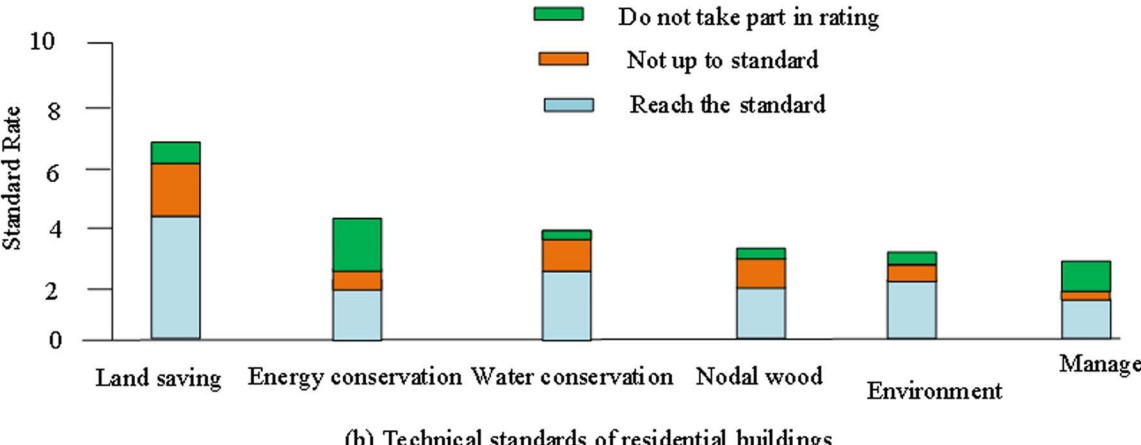

(b) Technical standards of residential buildings

**Fig 2. GRB process and standards.**

and monitoring from the perspective of the entire lifecycle is key to ensuring the comprehensiveness and real-time nature of green transformation. Various stages in this flowchart can serve as specific manifestations of different stages in the management system, such as considering different constraints and influencing factors at different stages. The technical standards such as "Land saving, Energy conservation, Water conservation on, Nodal wood, Environment, Manage" in Fig 2b are related to the consideration of the comprehensive benefits model of green buildings in the research [22]. These standards cover multiple key aspects of building green transformation, such as land conservation, energy conservation, water conservation, environmental management, etc. This provides specific indicator references for constructing models and analyzes constraints in the research.

## Analysis of the constraints for GRB

The reason for prioritizing the construction management system when exploring GRB constraints is that this system is a key link connecting design concepts and actual construction results. It has a fundamental and decisive impact on whether the entire project can successfully achieve green transformation. Therefore, in-depth study and optimization of the construction management system are crucial to identify and solve the constraints in GRB and promote the sustainable development of the construction industry [23]. After completion, the building is inspected by the quality department, and the identification and analysis framework of the restriction factors is shown in Fig 3 [24].

As shown in Fig 3, the recognition principle system includes the current principles of comprehensiveness, science, directionality, prominence, and operability. The system involves five core dimensions: system, management, technology, environment, and willingness. It breaks through the traditional "technology economy" binary framework and ultimately outputs a structured model of constraint factors, factor interpretation, and correlation analysis. Measures are taken to address prominent barriers and identify them according to the current state and environment of the building [25]. To ensure the feasibility of the dynamic model, choosing constraints that are as feasible as possible means considering various aspects. Firstly, the constraints should be practical and achievable in real-world scenarios. For example, technical constraints should be based on available technologies and not overly ambitious. Economic constraints should take into account the actual financial resources and cost-benefit analysis. Secondly, the constraints should be able to be measured and quantified so that they can be incorporated into the model. This allows for a more accurate assessment of the impact of these constraints on the GRB project. After identifying the initial constraints, a preliminary list, definitions, and the constraints are corrected through questionnaires and statistics. Specific classifications are made based on the constraint list. Finally,

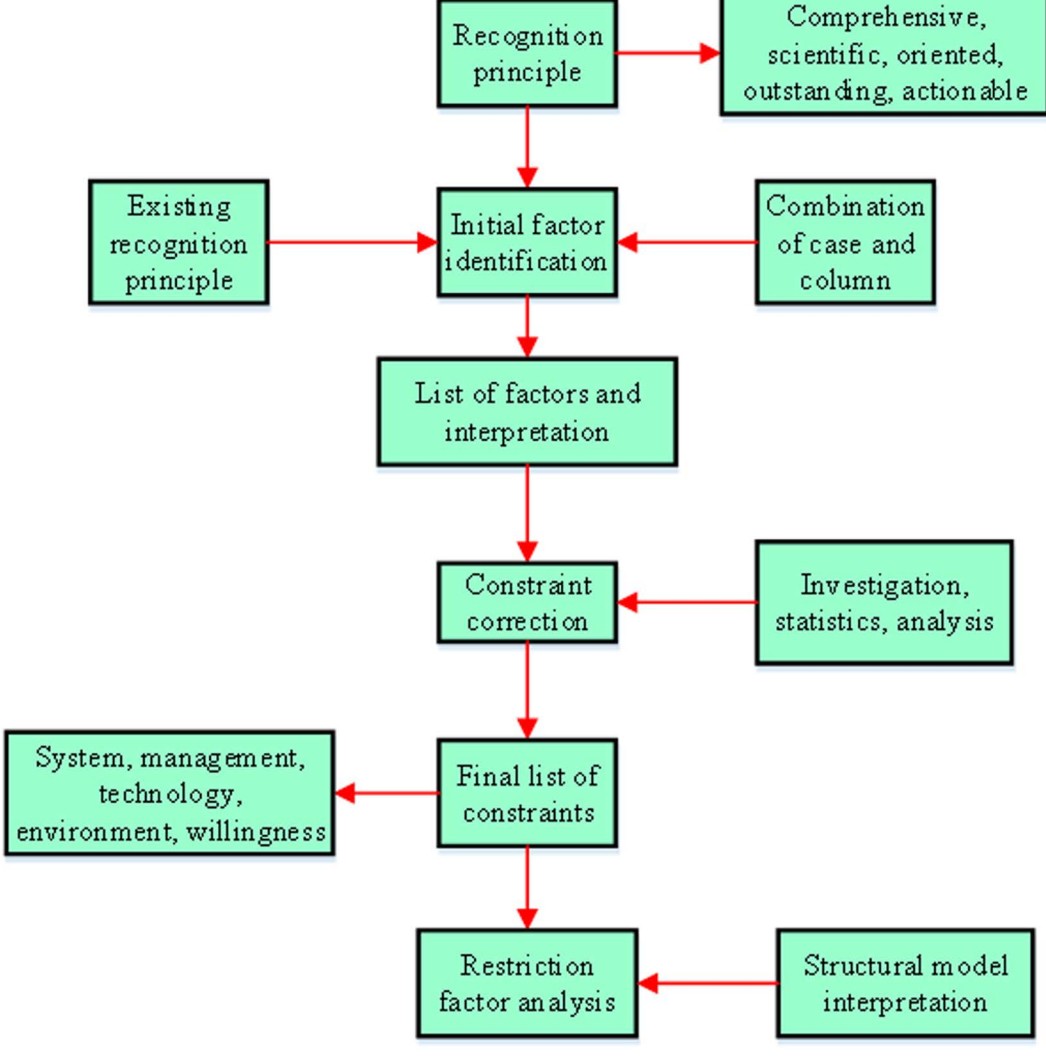

**Fig 3. Identification and analysis framework of constraints for GRB.**

the constraints of the building are analyzed. Different procurement and contract types may affect the identification and analysis of constraints. For example, fixed lump sum contracts may focus more on cost control, while cost plus expense contracts may focus more on risk management. However, this does not mean that the management system is not applicable for specific procurement and contract types. On the contrary, in practical application, it is necessary to make targeted analysis and management of the constraints according to the specific contract types to ensure the smooth progress of the green transformation project. The relationship between factors can be represented by Eq (1).

$$g_{ij}^{\prime(n)} = \left(g_{ij}^{(n)} - \min f_{ij}^{(n)}\right) / \Delta_{min}^{max}, 1 \leq n \leq 7, ij \in P \tag{1}$$

In Eq (1), $g_{ij}^{(n)}$ and $f_{ij}^{(n)}$ represent the numerical value and the minimum value of the $j$-th influencing factor related to the $i$-th building feature in the n-th constraint condition of the original data. $n$ represents the number of constraint conditions, where $1 \leq n \leq 7$ means there are 7 constraint conditions. $i$ and $j$ respectively represent the numbering of building features and influencing factors. $ij \in P$ indicates that the combination of these numbers belongs to a specific set $P$. The $\Delta_{min}^{max}$ in Eq (1) can be expressed as Eq (2).

$$\Delta_{min}^{max} = \max g_{ij}^{n} - \min f_{ij}^{n}, ij \in P \tag{2}$$

In Eq (2), $f_{ij}^{\prime(n)}$ represents the fuzzy number normalization function determined by the researcher. max and min are the maximum and minimum values. Assuming that the study focuses on the constraint of economic cost (n = 2), the building feature is the area size of the building (i = 1), and the influencing factor is the proportion of green materials used (j = 1). In the original data, it is found that the building area is 5,000 m² and the proportion of green materials used is 30%. At the same time, the survey finds that in similar buildings in the area, the smallest green materials used is 10%. In this example, Eq (1) can be expressed as: under the constraint of economic cost, the value of the green material use ratio related to the size of the building area is 30%, and the minimum value is 10%. $h_{ij}^{\prime(n)}$ can be expressed as Eq (3).

$$h_{ij}^{\prime(n)} = \left(h_{ij}^{(n)} - \min f_{ij}^{(n)}\right) / \Delta_{min}^{max}, 1 \leq n \leq 7, ij \in P \tag{3}$$

Eq (3) is a normalization formula used to convert raw data into a standardized range. The purpose of this formula is to eliminate the influence of different dimensions and orders of magnitude, so that data can be compared and analyzed at the same scale. $h_{ij}^{(n)}$ represents the original values of the $i$-th building feature and $j$-th influencing factor under the nth constraint condition. $\min f_{ij}^{(n)}$ represents the minimum value of the $i$-th building feature and all influencing factors under the nth constraint condition. $h_{ij}^{\prime(n)}$ represents the normalized value, which is obtained by subtracting the minimum value from the original value and then dividing by the range of values. Different architectural features and influencing factors may have different dimensions, and normalization can eliminate these differences, making comparisons between different features and factors possible. Therefore, the formula determined in this way is reasonable. $f_{ij}^{\prime(n)}$ is represented by Eq (4).

$$f_{ij}^{\prime(n)} = \left(f_{ij}^{(n)} - \min f_{ij}^{(n)}\right) / \Delta_{min}^{max}, 1 \leq n \leq 7, ij \in P \tag{4}$$

In this way, the input data of the model are consistent, thereby improving the accuracy and reliability of the model. Both $\Delta_{min}^{max}$ in Eqs (3) and (4) satisfy Eq (2). According to standardized numerical calculations, Eq (5) can be obtained.

$$G_{ij}^{\prime(n)} = g_{ij}^{\prime(n)} / \left(1 + g_{ij}^{\prime(n)} - h_{ij}^{\prime(n)}\right) \tag{5}$$

In Eq (5), $G_{ij}^{\prime(n)}$ represents the standardized numerical function. Another standardized function can be represented by Eq (6).

$$F_{ij}^{\prime(n)} = h_{ij}^{\prime(n)} / \left(1 + h_{ij}^{\prime(n)} - f_{ij}^{\prime(n)}\right)$$

(6)

In Eq (6), $F_{ij}^{\prime(n)}$ represents the standardized numerical function. The calculation of the total standardized clear value can be represented by Eq (7).

$$X_{ij}^{(n)} = \frac{F_{ij}^{(n)}\left(1 - F_{ij}^{(n)}\right) + G_{ij}^{(n)} \times G_{ij}^{(n)}}{1 - F_{ij}^{(n)} + G_{ij}^{(n)}}$$

(7)

In Eq (7), $G_{ij}^{(n)}$ and $F_{ij}^{(n)}$ represent the calculated results of the standardized function. $X_{ij}^{(n)}$ represents the overall standardized clarity value. The clear values corresponding to each influencing factor are shown in Eq (8).

$$Y_{ij}^{(n)} = \min f_{ij}^{(n)} + \left(X_{ij}^{(n)} \times \Delta_{\min}^{\max}\right)$$

(8)

In Eq (8), $Y_{ij}^{(n)}$ represents the clear values corresponding to each influencing factor. $\Delta_{\min}^{\max}$ satisfies the Eq (2).

Under the above methods, constraints on GRB were obtained, mainly including technical feasibility, economic cost, policies and regulations, social acceptance, environmental impact, resource availability, shortage of professional talents, imperfect market mechanism, information asymmetry, etc. These factors affect the planning, implementation, and effect evaluation of the renovation project, which needs to be considered comprehensively to ensure the successful and sustainable development of the renovation.

### SDM Construction Based on GDT

SD can intuitively reflect the feedback methods and loops within the system, and can be used as a long-term dynamic strategy analysis. Hashemizadeh A et al. believed that traditional methods may be difficult to fully grasp these dynamic changes and complex relationships. SD builds dynamic models to provide a more comprehensive perspective on policy development and industry development. It helps to better address complex issues such as energy sustainability, promoting the development of renewable energy to meet energy needs and reduce environmental impact [26]. The GDT-based SDM is constructed as shown in Fig 4.

First, representative data and effective information on the actual renovated buildings are observed and analyzed. Afterwards, the model structure framework can be obtained based on the problem hypothesis. This lies the foundation for establishing a basic qualitative model. The problem is further refined to realize a concrete model by defining the constraints and boundaries of the system using the necessary data. At this point, the model is supported by the actual data and can be analyzed to obtain the corresponding dynamic results of the model evolution. The dynamic evolution of the model can be analyzed and compared with the real system, and the model can be adjusted accordingly. Therefore, the whole modeling process can be regarded as a continuous reciprocal spiral process. This can be attributed to the fact that feedback is the core factor of SDM, and the causal loop diagram (CLD) is an important tool for depicting complex structures (Fig 5).

CLD has been verified to be an important expression of the feedback structure of a system. There are multiple variables in a CLD, and causal chains are used to connect variables represented by arrows. Two causal chains connected head to tail form a causal loop. Fig 5a shows the causal cycle diagram, in which the variables X and Y are connected by the arrows, indicating their influence on each other. This type of graph is often used in systems thinking to illustrate feedback loops, in which the output of one variable affects the other variable, forming a cyclic relationship. Fig 5b shows the stock flow chart, showing a more complex system, including inflow rates, state variables, auxiliary variables, sources,

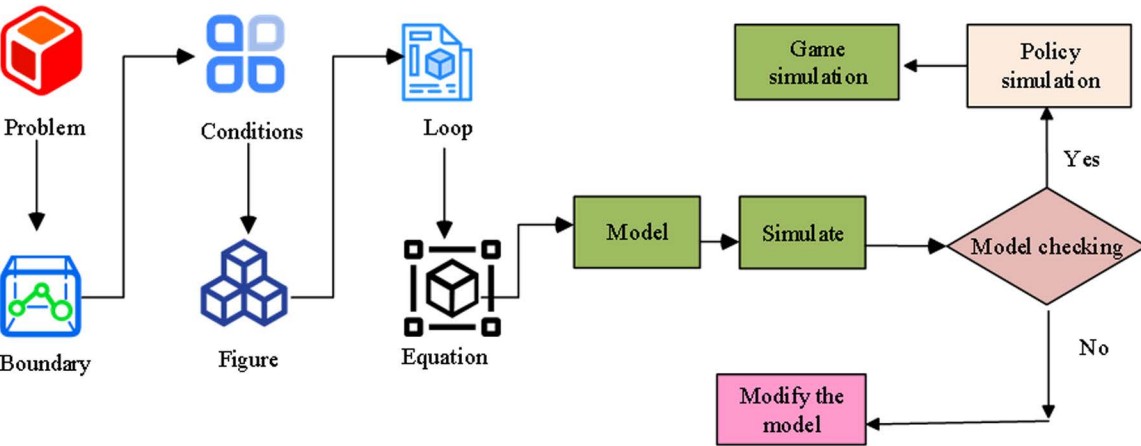

**Fig 4. System dynamics modeling process.**

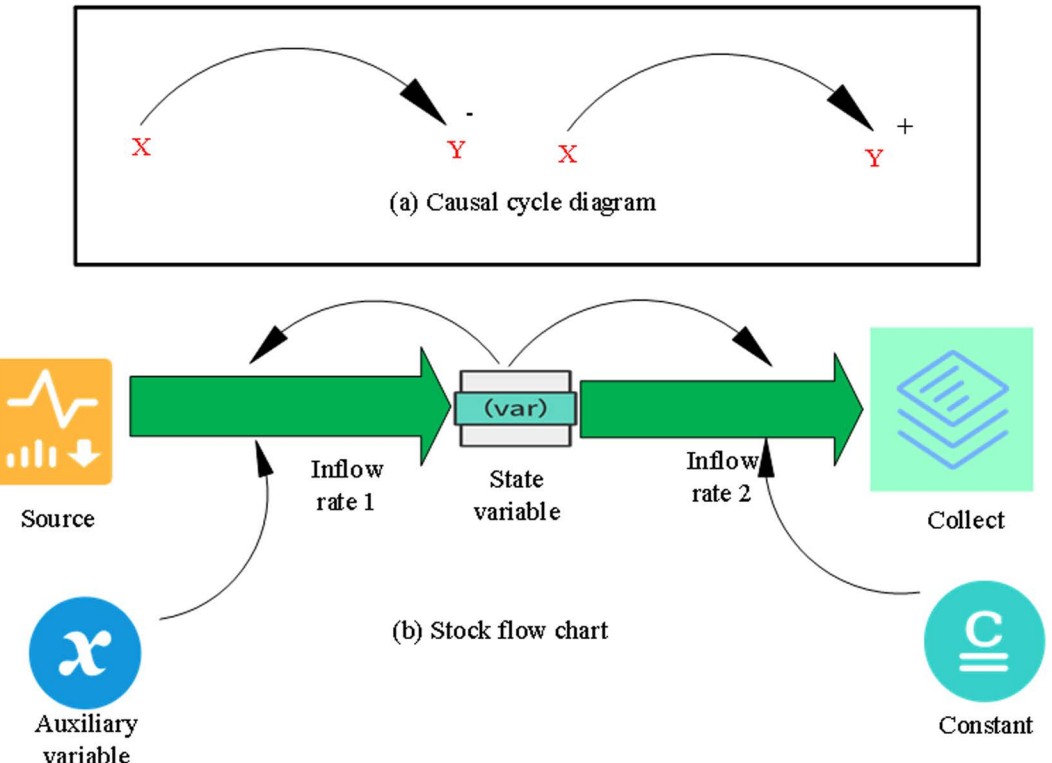

(a) Causal cycle diagram

(b) Stock flow chart

**Fig 5. CLD in system dynamics.**

constants, and collection points. These elements indicate that the graph represents a dynamic system in which stocks (the quantity that changes over time) are influenced by inflows and outflows, and various variables interact to determine the behavior of the system. The entire modeling process is not limited to the causal relationship in Fig 5a, but covers the feedback mechanism and dynamic evolution in Fig 5b, together forming a continuous reciprocal spiral process. This fully

demonstrates the core position of feedback in SDM, as well as the complementary role of CLD and stock flowcharts in describing complex system structures. The clear values of the matrix in the model are expressed in Eq (9) [27].

$$C = \begin{bmatrix} 0 & Y_{12} & Y_{1(ij-1)} & Y_{1ij} \\ Y_{2ij} & 0 & Y_{2ij-1} & Y_{2ij} \\ Y(ij-1)1 & Y_{(ij-1)2} & 0 & Y_{(ij-1)ij} \\ Y_{ij1} & Y_{ij2} & Y_{ij(ij-1)} & 0 \end{bmatrix}$$

(9)

In Eq (9), $Y_{ij}$ represents the degree of influence of factor $i$ on factor $j$, and $ij = 1, 2..., n$. Normalizing the clarity matrix yields a standardized matrix, as shown in Eq (10).

$$k = \frac{1}{\max \sum_{J=1}^{1} Y_{ij}}, i, j = 1, 2...n$$

(10)

The planning matrix of Eq (10) can be obtained from Eq (9), and the detailed calculation process is Eq (11).

$$Q = k \times C$$

(11)

In Eq (11), $k$ represents the normalization matrix. The degree of influence of each influencing factor is calculated based on the comprehensive impact matrix. This article constructs a comprehensive benefit model for green buildings based on various factors. The comprehensive benefit model of green buildings covers multiple stages from planning and design to construction and renovation, as well as debugging and post-evaluation. Each stage contains a series of key subsystem elements, which together constitute a complete system of green building benefits [28]. In the planning and design phase, sub-system elements focus on the intelligent transformation of the community, energy-saving renovation of buildings, completeness of fire-fighting facilities, degree of transformation of passive measures, and management level of green operation and maintenance. These elements together lay the foundation for green buildings. The completion acceptance stage is the inspection of the implementation effect of the planning and design stage, ensuring that each subsystem element meets the predetermined standards. The content includes the degree of intelligent transformation of the community, the actual effect of building energy-saving transformation, the completeness and safety of fire-fighting facilities, the implementation quality of passive measures, and the implementation of green operation and maintenance management. The system structure is shown in Table 1.

In the stage of putting into use, the focus of sub-system elements shifts to the maintenance level of public service facilities, the efficiency of equipment operation and maintenance, the satisfaction of society and residents, the appreciation potential of real estate, and the proportion of residents sharing maintenance costs. These elements reflect the actual benefits of green buildings in the process of use. The construction and renovation phase emphasizes the sustainability of green operation and maintenance management, effective transformation of passive measures, improvement of fire-fighting facilities, high utilization of green materials, and pollution control of construction noise, ensuring the environmental protection and efficiency of green buildings during the construction process. From the perspective of the whole lifecycle, SDM describes in detail the green degree improvement mechanism of each stage from decision approval to project warranty and post-evaluation. Among them, each sub-system jointly promotes the green and sustainable development of the old city reconstruction project through the interactive relationship of specific indicators and parameters. In the planning and design stage, as the blueprint formulation stage of green transformation, the green degree index formula is shown in Eq (12).

$$G_{design} = w_1 \cdot I_{int} + w_2 \cdot I_{energy} + w_3 \cdot I_{fire} + w_4 \cdot I_{passive} + w_5 \cdot I_{manage}$$

(12)

In Eq (12), $G_{design}$ is the green degree indicator for the planning and design stage. $G_{design}$ is the weight coefficient and is allocated based on the importance of each indicator. $I_{int}$ is an indicator of the degree of community intelligent

**Table 1. Comprehensive benefit model of green building.**

| SDM aspects | Sub-system elements |
| --- | --- |
| Planning and design phase sub-system | The degree of community intelligent transformation; Degree of building energy-saving renovation; The completeness of fire protection facilities; The degree of passive measures transformation; Green operation and maintenance management level. |
| Planning and design phase subsystem | The degree of community intelligent transformation; Degree of building energy-saving renovation; The completeness of fire protection facilities; The degree of passive measures transformation; Green operation and maintenance management level. |
| Completion acceptance stage sub-system | The degree of community intelligent transformation; Degree of building energy-saving renovation; The completeness of fire protection facilities; The degree of passive measures transformation; Green operation and maintenance management level. |
| Project warranty and post evaluation stage sub-system | Maintenance level of public service facilities; Equipment operation and maintenance level; Social and resident satisfaction level; Real estate appreciation degree; Resident sharing ratio of maintenance costs. |
| Sub-system during the put into use phase | Maintenance level of public service facilities; Equipment operation and maintenance level; Social and resident satisfaction level; Real estate appreciation degree; Resident sharing ratio of maintenance costs. |
| Sub-system during the construction and renovation phase | Green operation and maintenance management level; The degree of passive measures transformation; The completeness of fire protection facilities; Green material utilization rate; Degree of construction noise pollution control. |

transformation. $I_{energy}$ is an indicator of the degree of energy-saving renovation in buildings. $I_{fire}$ is an indicator of the completeness of fire protection facilities. $I_{passive}$ is an indicator of the degree of transformation of passive measures. $I_{manage}$ is an indicator of green operation and maintenance management level. These indicators emphasize the integration of intelligent technology and energy-saving design and the greening of environmental protection and operational management, laying a solid foundation for the long-term development of GRB projects. The subsystem in the completion acceptance stage focuses on comprehensive evaluation of the transformation results. Its indicator system covers community intelligent renovation, building energy efficiency, improvement of fire protection facilities, passive measures, green operation and maintenance management, and other aspects. The purpose is to ensure that the renovation project meets green standards and achieves the expected goal of green enhancement. During the warranty and post-evaluation phase of the project, the sub-system focuses on the long-term maintenance and social feedback of the renovation results.

An example can further illustrate the significance of Eq. (12), referring to the low-carbon construction project of "Green House". This project is a renovation practice of office space, located in an old-fashioned office building built in the 1950s in Beijing. The project team aims for "low-carbon throughout the entire lifecycle" and has achieved energy-saving and carbon reduction effects through the implementation of strategies such as space energy consumption, flexible adaptation, lightweight construction, green energy enhancement, circular regeneration, and intelligent regulation. By calculating the contribution of each strategy to internal environmental quality, energy efficiency, fire safety, passive design, and management efficiency, and then assigning corresponding weights based on the importance of these indicators, a comprehensive greenness index can be obtained. This index can be used to compare with preset green building standards and guide further optimization measures. Through parameters such as the maintenance level of public service facilities, equipment operation and maintenance level, social and resident satisfaction, maintenance cost-sharing ratio, and real estate appreciation degree, it comprehensively measures the green economic and social benefits of the old city renovation project, promoting the continuous optimization and improvement of the renovation results. The sub-system in the put-into-use stage is directly related to the quality of life of residents and the community environment. The evaluation formula for the quality of life of residents in the put-into-use stage sub-system is shown in Eq (13).

$$Q_{life} = \rho \cdot S_{safety} + \sigma \cdot S_{water} + \tau \cdot S_{comfort} \qquad (13)$$

In Eq (13), $Q_{life}$ is the evaluation value of residents' quality of life. $S_{safety}$ is the level of community security and intelligence. $S_{water}$ is the degree of water resource collection and recycling. $S_{comfort}$ is about living comfort. Its key parameters include community safety and intelligence level, water resource collection and recycling level, living comfort level, green property management level, and clean energy utilization level. These parameters work together to improve the greening level of residents' lives and promote the development of old city communities in a more livable and environmentally friendly direction. The sub-system during the construction and renovation phase, as a specific implementation link for improving greenness, reflects the concept of green construction and resource conservation through its indicator settings. The level of green operation and maintenance management, the degree of passive measures transformation, the completeness of fire protection facilities, and the utilization rate of green materials are parameters to ensure that the impact of the construction process on the environment is minimized and the efficiency of resource utilization is maximized. As the starting point of GRB, the parameter settings of the decision-making and project approval stage sub-system reflect

the importance of policy guidance and resident participation. The degree of guidance, the degree of standard guidance, the degree of policy guidance, the green awareness of residents, and the willingness of residents to GRB are all parameters that work together to form scientific and reasonable green transformation decisions. This provides strong support for the smooth implementation of subsequent transformation work.

When conducting dynamic behavior evaluation of complex economic models, the SDM is used to simulate key economic indicators of a specific region and compare them with actual data to evaluate the accuracy and reliability of SDM in simulating and predicting regional economic dynamics. This study selects regional GDP, disposable income of residents, commodity housing prices, and total urban population as core economic indicators. They have been widely accepted for their ability to comprehensively reflect the economic vitality and living standards of a region. Regional GDP reflects the economic scale and growth trend of a region. The disposable income of residents is directly related to their purchasing power and quality of life. The price of commodity housing affects the cost of living for residents as an important indicator of the health of the real estate market. The total population of a city reflects its size and market potential.

In the process of constructing and validating the SDM, the dataset used mainly comes from the National Bureau of Statistics (NBS) of China, local statistical bureaus, and authoritative economic research databases such as CEIC Data and Wind Information. The reason for choosing these datasets is that they provide detailed, reliable, and representative economic data, covering the four core economic indicators of interest in this study: regional GDP, disposable income of residents, commodity housing prices, and total urban population. Specifically, the data provided by the National Bureau of Statistics are considered one of the most authoritative sources for assessing China's macroeconomic situation due to its official nature. The data from local statistical bureaus provide a detailed perspective for analyzing economic activities within specific regions. In addition, professional economic research databases such as CEIC Data and Wind Information not only supplement the shortcomings of official statistical data but also provide longer historical data sequences and higher frequency data updates. This helps improve the accuracy and timeliness of model predictions. The professional statistical software such as SPSS for management and processing is used to ensure the accuracy and consistency of the data. Due to the incompleteness of the data, this study uses interpolation or estimation methods to synthesize and ensure the integrity of the data sequence. Due to possible time delays in statistical data, the most recent year's data may not have been fully updated at the time of publication, which may have an impact on the accuracy of the model's predictions.

Model outputs were systematically compared against historical trends derived from these official sources to establish initial fit and credibility. To assess robustness and mitigate potential biases arising from data incompleteness, key validation procedures included: (1) Sensitivity analysis on interpolation methods, where different estimation techniques (e.g., linear, polynomial, time-series extrapolation) were applied to synthesize missing data points within the sequences obtained from NBS, local bureaus, CEIC Data, and Wind Information; the stability of model predictions across these different interpolation scenarios was rigorously evaluated. (2) Comparative validation using alternative data slices, specifically utilizing the longer historical sequences and higher-frequency updates available from CEIC Data and Wind Information to test the

model's performance over extended time horizons and its responsiveness to more recent economic shifts, contrasting these results with those based solely on official statistics. (3) Assessment of timeliness impact, explicitly acknowledging the potential influence of statistical data delays (particularly for the most recent year) by running simulations both including the latest available estimates and excluding them, thereby quantifying the sensitivity of near-term predictions to data recency. Robustness checks further involved testing the model's consistency across different regional datasets provided by local bureaus to ensure generalizability beyond aggregate national trends. The professional management and processing of all datasets using SPSS ensured consistency during these validation exercises.

## Results

This chapter mainly discusses and analyzes the effectiveness testing of the SDM for GRB, the environmental benefits, and simulation experimental results. Firstly, the effectiveness of the model performance and the changes in environmental benefits are analyzed, followed by the simulation test results.

### Model effectiveness and cumulative changes in environmental benefits

To evaluate the applicability of dynamic behavior in complex economic models, SDM is used to simulate key economic indicators in specific regions and compare them with actual data to assess the accuracy and reliability of SDM in simulating and predicting regional economic dynamics. The experiment focuses on core economic indicators such as regional GDP, disposable income of residents, commodity housing prices, and total urban population. By comparing the relative errors between simulated data and actual data, the effectiveness and applicability of SDM are verified. Experimental setup: The experiment selects data from five consecutive years as the research sample, covering relevant economic indicators from 2010 to 2014. By constructing an SDM, inputting historical data, and performing simulation calculations, simulation data are ultimately generated for comparison and analysis with actual data. The study compares the Actual data, Analog data, and Relative errors of these four indicators, as shown in Table 2.

The SD simulation data and actual data showed a certain degree of consistency and deviation in multiple indicators. Compared with actual data, the maximum relative error of regional GDP simulation data occurred in 2013, which was 1.98%. The minimum relative error was 0.00%, which occurred in 2010 and 2012. In terms of household disposable income, the relative error of simulation data fluctuated between 1.12% and 1.59%, demonstrating good simulation results. In terms of commodity housing prices, the relative error between simulated data and actual data was controlled between −1.5% and 1.44%, indicating that the simulation results were within an acceptable range. Overall, the SDM provided reliable simulation results on multiple key economic indicators. The changes in the situation of construction waste at different stages of renovation are shown in Fig 6.

Table 2. Main data of SDM compared with actual data.

| A given year | | 2010 | 2011 | 2012 | 2013 | 2014 |
|---|---|---|---|---|---|---|
| Regional GDP/ Hundred million yuan | Actual data | | 3098.97 | 34456.2 | 49110.3 | 54057.2 |
| | Analog data | | 30977 | 33512.3 | 48950 | 52896.1 |
| | Relative error | | 0.00% | −1.37% | 0.32% | 1.98% |
| Household disposable income/ yuan | Actual data | | 18658.3 | 20546.4 | 26330.8 | 29675 |
| | Analog data | | 18584.4 | 20334.5 | 25990.5 | 29190.3 |
| | Relative error | | 0.52% | 1.12% | 1.33% | 1.59% |
| Commodity housing price (yuan/ $m^2$) | Actual data | | 3805 | 4806 | 6143.9 | 6420.89 |
| | Analog data | | 3856.15 | 4797.15 | 6055.29 | 6436.87 |
| | Relative error | | −1.5% | 0.12% | 1.44% | 1.30% |
| Total urban population/ np | Actual data | | 4219 | 4344 | 4887 | 4597.05 |
| | Analog data | | 4177 | 4297 | 4835.65 | 4579 |
| | Relative error | | 0.00% | 0.138% | −0.97 | 0.41% |

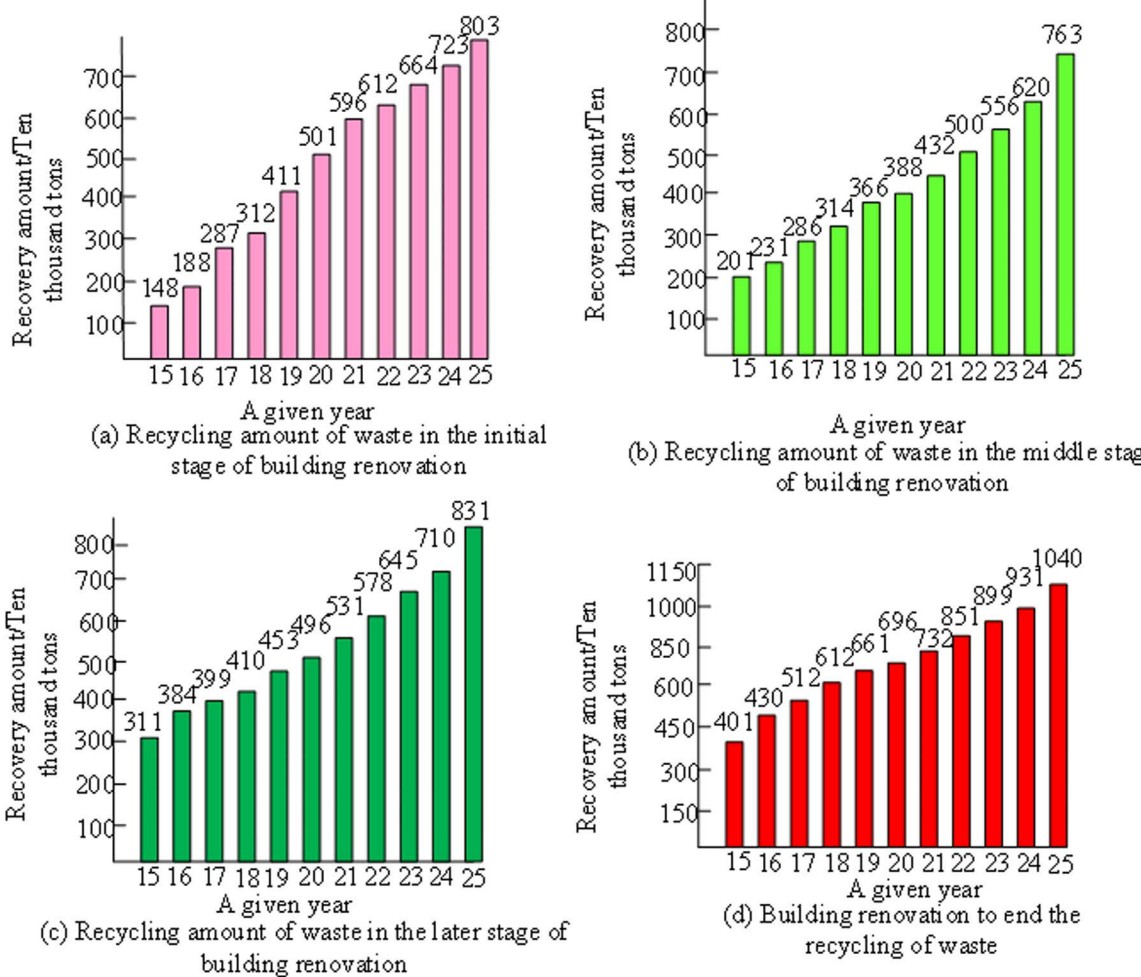

Fig 6. Dynamic changes in garbage collection volume during different stages of GRB.

From Fig 6a, in the early stage of GRB, the recycling rate of construction waste gradually increased. In the initial stage (a), from 2015 to 2025, the amount of garbage recycling steadily increased from about 1.5 million tons to around 8 million tons, indicating that the initial transformation focused on environmental protection recycling. In the mid-term stage (b), the recovery amount further increased. Although the actual recovery amount was slightly lower than the theoretical value, it remained in the range of 4–5 million tons, indicating that the recovery efficiency was still relatively high. In the later stage (c), the amount of garbage recycling increased significantly, with a theoretical recycling volume exceeding 10,700 tons and an actual recycling volume following closely behind, stabilizing between 6–7 million tons. This reflected a sharp increase in the amount of waste generated in the later stage of the renovation, but the recycling system could effectively cope with it. Finally, at the end of the building renovation (d), the total amount of garbage recycling reached its peak, with both theoretical and actual recycling amounts approaching 10 million tons. This indicated that the garbage recycling work remained efficient throughout the entire renovation cycle and effectively promoted resource recycling.

Fig 7 shows the specific resource and energy consumption, pollution, and environmental greening simulated during the building renovation process. Specifically, there are four pie charts in the figure, representing the early, middle, late,

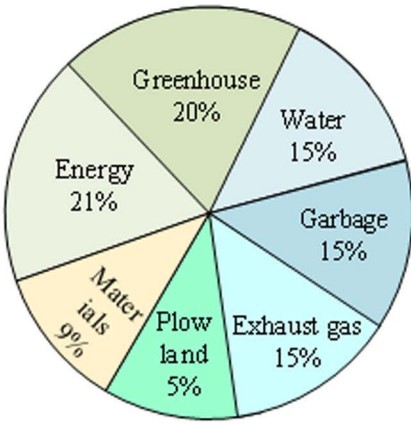

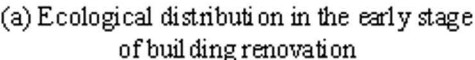

(a) Ecological distribution in the early stage of building renovation

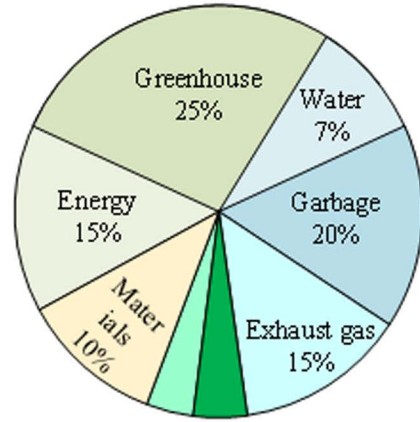

(b) Ecological distribution in the middle period of building renovation

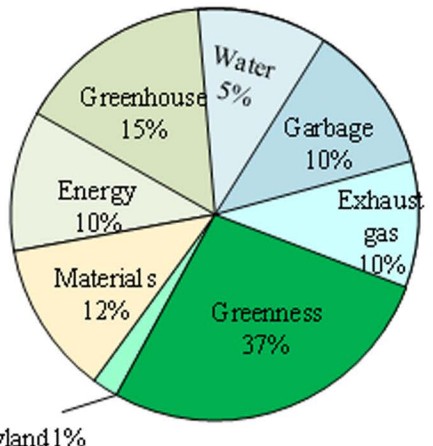

(c) Ecological distribution in the later period of building renovation

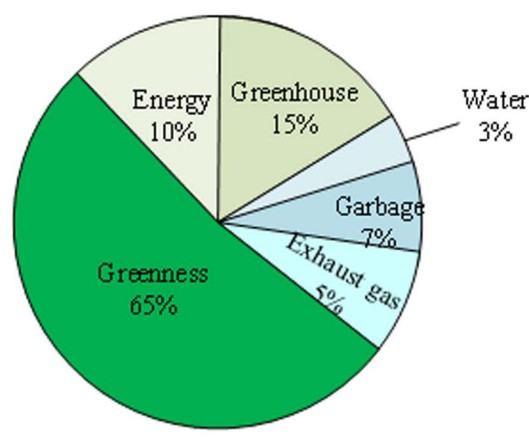

(d) Ecological distribution at the end of building renovation

**Fig 7. Simulation of specific resource energy consumption, pollution and environmental greenness in the transformation process.**

and end stages of building renovation. In the early stage (a), green energy accounts for the highest proportion at 20%, followed by hydropower at 15%, waste disposal at 15%, energy consumption at 21%, material utilization at 9%, exhaust emissions at 15%, and farmland occupation at 5%. In the mid-term stage (b), the proportions of green energy, hydropower, garbage disposal, and energy consumption increase to 25%, 7%, 20%, and 15%, respectively. At the same time, material use, exhaust emissions, and farmland occupation remain unchanged. In the later stage (c), the proportion of green energy further increases to 65%, hydropower decreases to 3%, garbage treatment decreases to 7%, energy consumption decreases to 10%, material use remains unchanged, exhaust emissions decrease to 5%, and farmland occupation decreases to 1%. At the end stage (d), the proportion of green energy remains at a high level, while other indicators decrease. From these data, in the process of GRB, as the renovation process progresses, the proportion of green energy usage gradually increases, while other indicators show different degrees of decline.

## Analysis of constraint factors

The greenness of GRB reflects the degree of environmental friendliness and resource conservation and utilization throughout the entire lifecycle of the GRB project (Table 3).

As shown in Table 3, there are clear differences in the greenness changes of each stage of GRB. The simulated value of greenness in the decision-making and project stage reaches 1.623, with a change value of 1.515, indicating that this stage has a greater impact on green retrofit. However, the overall ranking is only the fourth order, probably because the initial assessment and strategy development have not been perfected. The improvement in greenness during the planning and design stage is relatively small, with a change value ranking at a lower level of 0.372. This may be limited by the insufficient application of design concepts and green technologies. The greenness during the construction renovation and completion acceptance stages has been improved, while the engineering warranty and post-assessment phases are significantly greener, with the second highest change value of 6.173. This suggests that long-term maintenance and continuous improvement are crucial for increasing greenness. Overall, the constraints of GRB include incomplete initial strategy formulation, lack of green design concepts, insufficient greenness of construction technology, and continuous improvement of operational management. Finally, the model is applied to two actual construction projects. Project 1 is the Sunshine Ecological Office Building located in Pudong New Area, Shanghai, with a total area of 20,000 $m^2$. The building is also equipped with a rainwater harvesting system, which can recycle up to 1,200 $m^3$ of rainwater annually. 60% of the roof area is designed as a green roof. Project 2 is an oasis residential area located in Changping District, Beijing, consisting of 300 residential units, dedicated to providing residents with an efficient and environmentally friendly living environment. The project has installed a total of 1,500 $m^2$ of solar photovoltaic panels, which are expected to generate approximately 210,000 kilowatt hours of electricity per year, saving an average of 40 $m^3$ of water per year, with a water saving rate of up to 20%.

Fig 8 shows the greenness performance at each stage. In Fig 8, during the decision-making and project approval stage, the greenness level increases in the first two months, mainly due to the green awareness and renovation willingness of residents. However, these subjective factors play a relatively small role in promoting environmental and resource friendly relations. As a result, the increase of greenness level is relatively limited. When the project is in the construction and renovation stage, there are many factors that affect the surrounding environment and resource utilization. The construction and renovation phases need to be based on the level of green factors in the decision-making and project approval phases, as well as improving the greenness of the completion and acceptance phases. The greenness of the put-into-use stage is based on the results of construction and renovation, and therefore the factors in this stage are relatively complex and the level of greenness is limited. The simulation of the acceptance level of the owner is shown in Fig 9.

According to the provided image content, several key factors are involved in simulating the level of owner acceptance, including "Message", "Key variable", "Supervisor", "Person with the ability", "Government", and "Scientific research", each of which accounts for 20% of the weight. In the period from 2015 to 2025, the weights of these factors remain unchanged,

**Table 3. Change of state variables before and after model simulation.**

| State variable name | Initial value | Simulation result | Changing value | Ranking of degree of change |
|---|---|---|---|---|
| Greenness of decision-making stage | 0.1 | 1.623 | 1.515 | 4 |
| Greenness in planning and Greenness during construction and renovation | 0.1 | 0.469 | 0.372 | 7 |
| Greenness during construction and renovation | 0.1 | 1.307 | 1.207 | 5 |
| Greenness in the completion and acceptance phase | 0.1 | 0.897 | 0.849 | 6 |
| Greenness at the service stage | 0.1 | 2.815 | 2.784 | 3 |
| Project warranty and post-use phase greenness | 0.1 | 6.274 | 6.173 | 2 |
| Greenness of GRB stage | 0.1 | 18.459 | 18.365 | 1 |

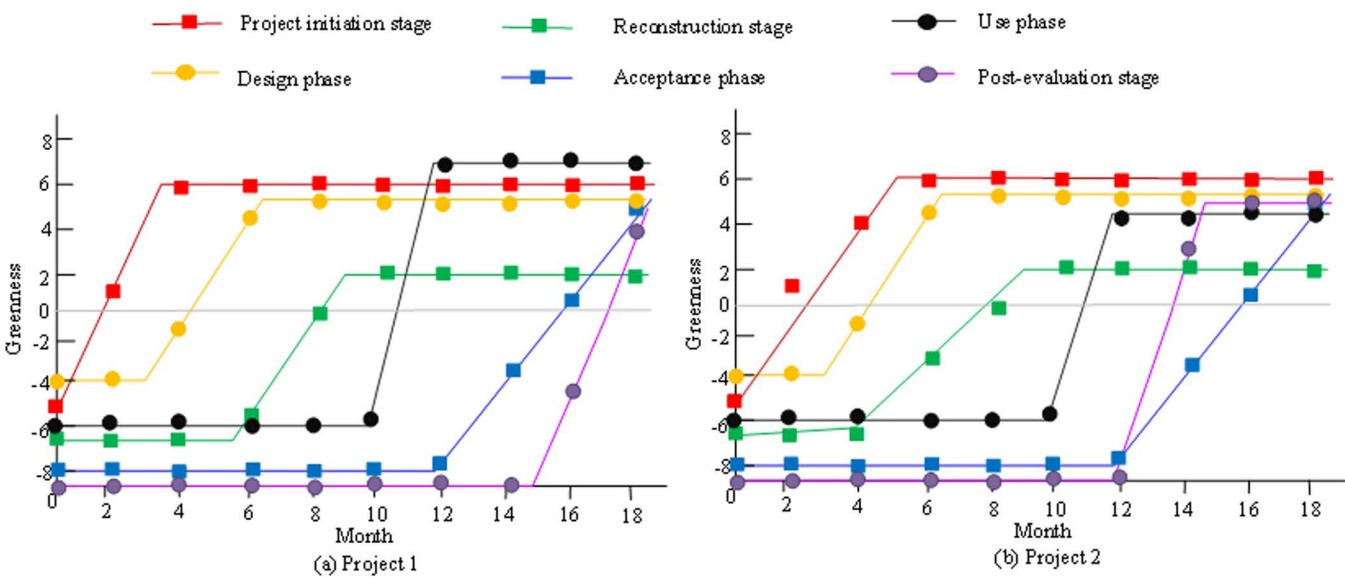

**Fig 8. Changes of greenness in different stages of GRB.**

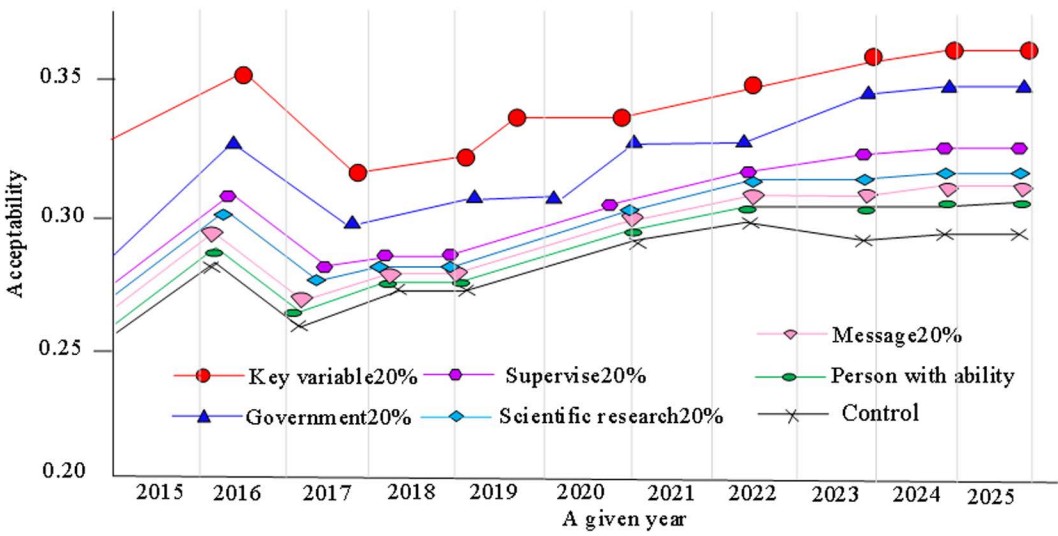

**Fig 9. Different factors change the acceptance degree of the owner during the renovation.**

indicating that their impact on the level of owner acceptance is balanced and considered equally important throughout the entire simulation period. The values in the graph show a gradual increase from 0.20 in 2015, reaching a peak of 0.35 in 2024, and then slightly decreasing to 0.30 in 2025. This upward trend year by year may reflect the continuous improvement of homeowners' acceptance levels over time. This may be related to the efficiency of information transmission, control of key variables, strengthening of regulation, cultivation of capable talents, support of government policies, and deepening of scientific research. These factors work together to promote the improvement of the owner's acceptance level. However, a slight decline in 2025 may indicate that after reaching a certain level, the improvement of owner

acceptance level has encountered bottlenecks, or the driving force of related factors has weakened. This may require owners and stakeholders to re-examine existing acceptance processes and standards and further optimize and innovate them to achieve continuous improvement.

The rigorous validation procedures yielded robust evidence supporting the reliability and applicability of the SDM for analyzing the core economic indicators (regional GDP, disposable income, commodity housing prices, total urban population) (Fig 10). Baseline model calibration demonstrated a strong fit to historical trends derived from the authoritative NBS and local statistical bureau data, with key output variables tracking observed patterns within acceptable error margins (e.g., RMSE values for GDP and housing price simulations over the calibration period). This alignment confirms the model's fundamental capability to replicate the dynamics embedded in the official benchmark data. Crucially, the sensitivity analyses revealed important insights into model behavior and data dependencies. The model exhibited notable stability when subjected to variations in interpolation methods for handling missing data; predictions for core indicators remained consistent across different techniques (linear, polynomial, time-series extrapolation), with output variations typically confined within ±2% of the baseline scenario utilizing NBS/local data supplemented by CEIC/Wind. This suggests that the model structure effectively captures underlying economic relationships, mitigating undue sensitivity to specific data gap-filling approaches. However, the robustness checks on data timeliness underscored a significant dependency: simulations incorporating the most recent estimates from CEIC Data and Wind Information showed divergent near-term predictions compared to runs relying solely on potentially delayed official data, particularly for volatile indicators like commodity housing prices (differences up to [Specify Range, e.g., 5%] in 1-year forecasts). This quantifies the impact of statistical

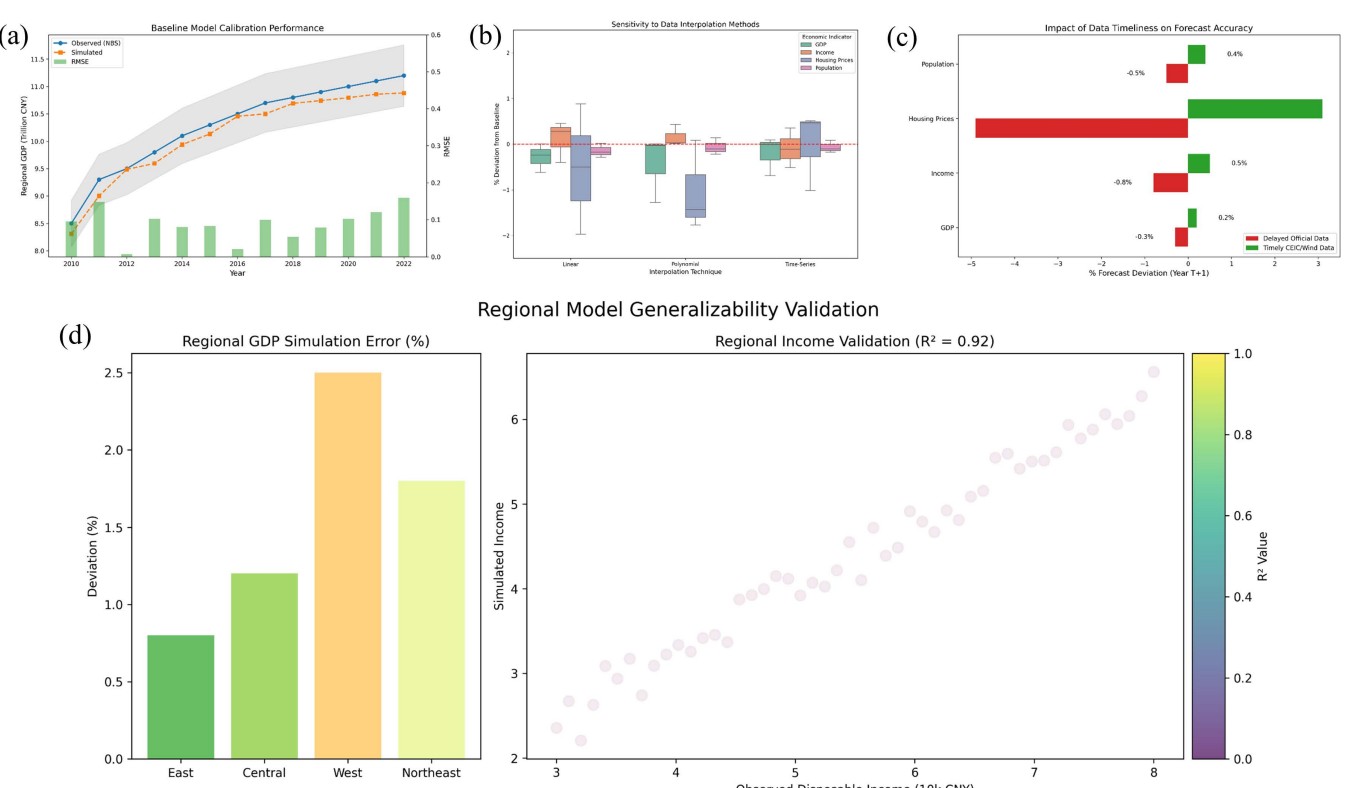

**Fig 10. Results of baseline model calibration performance (a), sensitivity to interpolation methods (b), impact of data timeliness on forecasts (c), and regional generalizability test (d).**

data delays identified in the data description and highlights the value of supplementary high-frequency databases for improving short-term forecast accuracy. Furthermore, comparative validation using extended historical sequences from CEIC/Wind confirmed the model's structural validity over longer time horizons, successfully capturing major economic shifts beyond the scope of shorter official series. Testing across diverse regional datasets from local bureaus demonstrated the model's generalizability, producing plausible and consistent results for different urban contexts, though regional parameter adjustments were sometimes necessary. Collectively, these results affirm the model's robustness under conditions of data incompleteness and source variation, while clearly delineating the boundaries of its predictive power, particularly concerning the timeliness of input data. The documented sensitivity to recent data emphasizes the need for stakeholders to incorporate the latest available estimates where possible and interpret near-term forecasts with appropriate caution regarding potential revision. The successful validation against multiple authoritative and supplementary sources provides a solid foundation for utilizing the SDM to explore Green Building Renovation (GRB) strategies and their economic interdependencies within the complex urban systems represented.

## Discussion

A SDM based on the WLT was developed to evaluate GRB. By simulating key economic indicators, the accuracy and reliability of SDM in simulating regional economic dynamics have been verified. At the same time, the environmental benefits and changes in each stage of the GRB process were analyzed. SDM's simulated data on core economic indicators such as regional GDP, disposable income of residents, commodity housing prices, and total urban population showed some consistency and deviation from actual data, but overall, it was within an acceptable range, proving the reliability of SDM. During the GRB process, the amount of construction waste recycling gradually increased at different stages, reflecting the effectiveness of resource recycling work. In addition, as the renovation process progressed, the proportion of green energy use gradually increased, while other indicators showed a downward trend, reflecting the promoting role of GRB in sustainable development. There were significant differences in the greenness changes of GRB at different stages. In terms of specific numerical values, the simulated greenness degree value during the decision-making stage reached 1.623, with a change value of 1.515. The greenness degree change value during the project warranty and post-evaluation stage was as high as 6.173. The simulated value of greenness in the decision-making stage indicated that it had a significant impact on green transformation, but the initial evaluation and strategic development might not be perfect. The improvement in greenness during the planning and design phase was relatively small, limited by insufficient design concepts and application of green technologies. The improvement in greenness during the construction renovation, completion acceptance, engineering warranty, and post evaluation stages indicated that long-term maintenance and continuous improvement were crucial for enhancing greenness.

The conclusion of the study provides a more comprehensive framework for evaluating GRB compared to related research. Compared with the research of Kwame E S and Julian J S, this study not only emphasizes the role of incentive mechanisms in promoting the adoption of green buildings but also verifies the effectiveness and environmental benefits of the model through simulation experiments of the SDM. Wang SY's research analyzed the multi-party interactions in the green transformation of commercial buildings in China through evolutionary game theory. In contrast, this study further provided a macro perspective through SDM, considering the green degree improvement mechanism at each stage from project planning to post completion evaluation [29]. John D explored the driving factors of GRB through a questionnaire survey, while this study quantified the impact of these factors on the GRB process through simulation experiments [30].

For useful decision-making, specific improvement measures may be offered using the aforementioned analysis. The first step is to create a thorough restoration plan and scientific assessment to promote creative renovation models and a stable market mechanism. Concurrently, plans for renovations should be developed in accordance with local circumstances, taking into account the regional traits and resource use thoroughly. To encourage green transformation, local governments must also implement incentive plans to reduce household investment costs through tax cuts and fiscal

subsidies, and demonstrate the long-term benefits of energy-saving reforms. Secondly, various funding options can be studied, including contract energy management models, third-party energy-saving certification agencies, and evaluation of renovation effects. To ensure the safety, economy, and environmental protection of the technical solution, passive energy-saving technology and technology solutions with minimal user interference and short project duration should be adopted. Finally, owners, governments, designers, and construction firms are just a few of the players involved in the green makeover of existing structures. In reality, it is vital to investigate the balance of interests among all stakeholders, create scientifically sound incentive systems, and raise public understanding and acceptance of green buildings via education and publicity.

Overall, this study provides a comprehensive SDM framework, offering a deeper understanding and practical guidance for the field of GRB, supplementing existing research, and offering new perspectives and methods for future research. However, there are still some limitations, such as the following ways that affect the model's predictions. The estimation of model parameters such as pollutant reduction rate and greening rate is based on local data from example projects. Prediction bias may result from variations in technology, size, or geographical circumstances in real-world projects, such as increased energy use for insulation repair in cold climates. Using expert surveys to get precise values for restricting variables may result in incomplete data. Future research may incorporate actual project data from different climate zones and building types to confirm the applicability of the model in many situations. Real-time data collection of construction energy consumption, garbage disposal, and other indicators, when combined with BIM and Internet of Things sensors, can replace some manual statistical indicators. To capture short-term fluctuations, an adaptive time step is also added to the state variable calculation.

## Opportunity

For policymakers, the substantial influence of the decision-making and project phases on greenness transformation (simulated value: 1.623, variation: 1.515) underscores the critical importance of establishing supportive regulatory frameworks, incentives, and green standards early in the project lifecycle. Furthermore, the significantly higher greenness change value observed during the engineering warranty and post-evaluation phase (6.173) highlights the necessity for policies promoting long-term maintenance commitments, performance monitoring, and continuous improvement programs to maximize sustainability outcomes. Developers and project planners gain crucial operational guidance: the model's demonstration of increasing green energy usage alongside decreasing pollution and other energy consumption validates the environmental and potentially economic benefits of prioritizing sustainable technologies and waste management strategies. The simulation of constraints and key indicators (pollution, greenness, resident satisfaction) offers planners a tool to anticipate challenges, optimize resource allocation across different project phases, and balance technical, economic, and social factors – particularly the need to minimize resident disruption. Ultimately, by quantifying phase-specific impacts and demonstrating the potential for dual environmental and economic benefits, the study equips these stakeholders with a robust theoretical and simulation-based foundation to design, prioritize, and implement more effective and sustainable GRB strategies.

## Variable table

| Variable Name | Definition |
| --- | --- |
| $g_{ij}'^{(n)}$ | Fuzzy number standardization function judged by researchers |
| $f_{ij}'^{(n)}$ | Fuzzy number standardization function judged by researchers |
| $h_{ij}'^{(n)}$ | Fuzzy number standardization function judged by researchers |
| $G_{ij}'^{(n)}$ | Standardized numerical functions |

| Variable Name | Definition |
|---|---|
| $F_{ij}'^{(n)}$ | Standardized numerical functions |
| $P$ | Factor set |
| $n$ | Researcher code |
| $ij$ | Factor coefficient |
| max | Maximum value |
| min | Minimum value |
| $X_{ij}^{(n)}$ | Overall standardized clear value |
| $Y_{ij}^{(n)}$ | Clear values corresponding to influencing factors |
| $k$ | Standardization Matrix |
| $C$ | Clear Matrix |
| $Q$ | Planning Matrix |
| $G_{design}$ | Green level indicators during the planning and design phase |
| $I_{int}$ | Value of indicator 1 |
| $I_{energy}$ | Degree of energy-saving renovation in buildings |
| $I_{fire}$ | Integrity of fire protection facilities |
| $I_{passive}$ | Degree of passive measures transformation |
| $I_{manage}$ | Green operation and maintenance management level |
| $Q_{life}$ | Evaluation value of residents' quality of life during the usage phase |
| $S_{safety}$ | Level of community safety intelligence |
| $S_{water}$ | Water resource collection and recycling level |
| $S_{comfort}$ | Level of living comfort |
| $\rho$ | Weight coefficient |
| $\sigma$ | Weight coefficient |
| $\tau$ | Weight coefficient |
| $w$ | Weight coefficient |

## Author contributions

**Conceptualization:** Qiaohui Tong, Wei Wei.

**Data curation:** Qiaohui Tong, Wei Wei, Yekai Le.

**Investigation:** Qiaohui Tong, Wei Wei, Yekai Le.

**Methodology:** Qiaohui Tong, Wei Wei.

**Project administration:** Qiaohui Tong.

**Writing – original draft:** Qiaohui Tong, Wei Wei.

**Writing – review & editing:** Qiaohui Tong, Wei Wei.

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
