## [Decision Letter · Decision Letter 0]

PONE-D-25-06830Constraints on Green Renovation of Buildings Based on the Theory of Whole Lifecycle and Green DevelopmentPLOS ONE

Dear Dr. Wei,

Thank you for submitting your manuscript to PLOS ONE. After careful consideration, we feel that it has merit but does not fully meet PLOS ONE’s publication criteria as it currently stands. Therefore, we invite you to submit a revised version of the manuscript that addresses the points raised during the review process. When revising your manuscript, please consider all issues mentioned in the reviewers' comments carefully: please outline every change made in response to their comments and provide suitable rebuttals for any comments not addressed. Please note that your revised submission may need to be re-reviewed. Please submit your revised manuscript by May 07 2025 11:59PM. If you will need more time than this to complete your revisions, please reply to this message or contact the journal office at plosone@plos.org . Please include the following items when submitting your revised manuscript:

We look forward to receiving your revised manuscript.

Kind regards,

Genyu Xu, Ph.D.

Academic Editor

PLOS ONE

Journal Requirements:

4. In the online submission form, you indicated that data will be available on reasonable request from the corresponding author.

Reviewers' comments:

Reviewer's Responses to Questions

**Comments to the Author**

1. Is the manuscript technically sound, and do the data support the conclusions?

Reviewer #1: Yes

Reviewer #2: Yes

2. Has the statistical analysis been performed appropriately and rigorously? 

Reviewer #1: Yes

Reviewer #2: Yes

3. Have the authors made all data underlying the findings in their manuscript fully available?

Reviewer #1: Yes

Reviewer #2: Yes

4. Is the manuscript presented in an intelligible fashion and written in standard English?

Reviewer #1: Yes

Reviewer #2: Yes

5. Review Comments to the Author

Reviewer #1: This paper proposes a system dynamics-based model for evaluating the comprehensive benefits of green building retrofits, analyzing the constraints and limitations faced during the retrofit process. The study, grounded in the theories of life cycle assessment and green development, simulates key indicators at various stages of green building retrofits, including pollution, greening, energy consumption, and resident satisfaction. It also emphasizes the significant role of engineering warranties and post-assessment phases in enhancing the greening level of buildings. The research holds certain value.

1.Analysis diagrams such as Fig. 1, "Management organization system for GRB based on WLT," which organizes various roles and responsibilities involved in green building retrofit projects through conceptual analysis and transformation history of GRB, and proposes a management organization system for GRB based on WLT, lack clear explanation of their specific literature sources and basis.

2.It is recommended to elaborate more on the potential impact of these estimation methods on the model's predictions in the discussion section and suggest ways to improve data collection and processing in future research. Future studies could consider incorporating data from more actual projects to validate the model's predictive capabilities, especially when applied to different regions and types of building projects.

3.Although the conclusion section summarizes the main points, it does not delve deeply into future research directions and practical application challenges of the method. It is suggested that the authors provide more analysis on potential practical issues and propose corresponding solutions, making the conclusions more forward-looking and practically guiding.

4.It is suggested to further supplement relevant references.

5.The format of the references is inconsistent.

6.Please confirm whether there are redundant and unnecessary icons in Figure 3.Meanwhile, there are a few layout issues in Figure 3, such as insufficient border height.

Reviewer #2: This manuscript explores constraints on green renovation of buildings (GRB) using a System Dynamics Model (SDM) based on Whole Lifecycle Theory (WLT) and Green Development Theory (GDT). The topic is timely and relevant, and the proposed framework offers insights useful for advancing sustainable urban development.

- Integrating WLT and GDT within an SDM framework is innovative and provides a clear theoretical foundation.

- The analysis effectively considers multiple phases of the renovation process, highlighting key factors that influence sustainability and environmental performance.

- The simulation results are informative and adequately validated with real-world economic and demographic data, enhancing the robustness and relevance of the findings.

Suggested areas for improvement include:

- Some parts of the manuscript could benefit from improved clarity and flow. A careful revision focusing on concise and straightforward sentence structure would enhance overall readability and accessibility.

- Clearer explanations of the variables and parameters used in equations would make the modeling approach easier to follow. Including a summary table clearly defining these variables would be helpful for readers.

- Enhancing figures with more detailed labels and descriptive captions would assist readers in interpreting the results. Additionally, clearer subheadings within the methodology and results sections could guide readers through the modeling process more effectively.

- Expanding the discussion of limitations to consider the role of assumptions, data estimation methods, and the potential generalizability of the model to different contexts could help clarify the scope and applicability of the findings.

- It would be beneficial to explicitly discuss how the findings could inform practical decisions, such as planning, policy-making, investment choices, or stakeholder coordination in green renovation efforts.

Overall, this manuscript addresses a meaningful issue and presents a sound modeling approach. With careful attention to language clarity, detailed explanation of variables, improved figure presentation, and enhanced practical implications, this work can contribute to research on sustainable building renovation.

6. PLOS authors have the option to publish the peer review history of their article (what does this mean? ). If published, this will include your full peer review and any attached files.

**Do you want your identity to be public for this peer review?** For information about this choice, including consent withdrawal, please see our Privacy Policy .

Reviewer #1: **Yes: ** Di-fei Zhao

Reviewer #2: No

---

## [Author Response · Author response to Decision Letter 1]

23 Apr 2025

PONE-D-25-06830

PLOS ONE

We appreciate the opportunity to revise our manuscript titled " Constraints on Green Renovation of Buildings Based on the Theory of Whole Lifecycle and Green Development " and are grateful for the insightful comments provided by the reviewers. These comments are valuable and very professional for revising and modifying our paper, as well as the important guiding significance to our researches. In the resubmission, we have provided detailed responses to the reviewers' comments. Revised portion are marked in red in the revised manuscript. We have tried our best to make all the revisions clear, and we hope that the revised manuscript meets the requirements for publication. Here are the point-by point responses:

Reviewer #1: This paper proposes a system dynamics-based model for evaluating the comprehensive benefits of green building retrofits, analyzing the constraints and limitations faced during the retrofit process. The study, grounded in the theories of life cycle assessment and green development, simulates key indicators at various stages of green building retrofits, including pollution, greening, energy consumption, and resident satisfaction. It also emphasizes the significant role of engineering warranties and post-assessment phases in enhancing the greening level of buildings. The research holds certain value.

Reply: Thank you very much for your affirmation and evaluation of the manuscript.

1.Analysis diagrams such as Fig. 1, "Management organization system for GRB based on WLT," which organizes various roles and responsibilities involved in green building retrofit projects through conceptual analysis and transformation history of GRB, and proposes a management organization system for GRB based on WLT, lack clear explanation of their specific literature sources and basis.

Reply: Thank you very much for your valuable feedback on the manuscript. The literature source and basis for Figure 1 have been revised as follows.

The WLT has a central place in the GRB system, stressing the need of auditing and monitoring at every phase, from project design and development to construction and settlement. The thoroughness and real-time nature of GRB may be guaranteed by developing a system based on the whole lifespan. Through the conceptual analysis and transformation history of GRB, the various roles and responsibilities involved in the GRB project are sorted out, thereby constructing the management organization system. The GRB management organization system based on WLT is proposed, as shown in Fig. 1 [19].

19.Aly IA, Kurttisi A, Dogan KM. Resilient coordination of multi-agent systems in the presence of unknown heterogeneous actuation efficiency and coupled dynamics: distributed approaches. International Journal of Systems Science, 2024, 55(9):1924-1946.

The WLT has a central place in the GRB system, stressing the need of auditing and monitoring at every phase, from project design and development to construction and settlement. The thoroughness and real-time nature of GRB may be guaranteed by developing a system based on the whole lifespan.

2.It is recommended to elaborate more on the potential impact of these estimation methods on the model's predictions in the discussion section and suggest ways to improve data collection and processing in future research. Future studies could consider incorporating data from more actual projects to validate the model's predictive capabilities, especially when applied to different regions and types of building projects.

Reply: Thank you for your constructive feedback on the manuscript. Under your guidance, the potential impact of the estimation method on model predictions and future research have been revised as follows:

Although this study has achieved some success, there are still some limitations, such as the following ways that affect the model's predictions. The estimation of model parameters such as pollutant reduction rate and greening rate is based on local data from example projects. Prediction bias may result from variations in technology, size, or geographical circumstances in real-world projects, such as increased energy use for insulation repair in cold climates. Using expert surveys to get precise values for restricting variables may result in incomplete data. Future research may incorporate actual project data from different climate zones and building types to confirm the applicability of the model in many situations. Real-time data collection of construction energy consumption, garbage disposal, and other indicators, when combined with BIM and Internet of Things sensors, can replace some manual statistical indicators. To capture short-term fluctuations, an adaptive time step is also added to the state variable calculation.

3.Although the conclusion section summarizes the main points, it does not delve deeply into future research directions and practical application challenges of the method. It is suggested that the authors provide more analysis on potential practical issues and propose corresponding solutions, making the conclusions more forward-looking and practically guiding.

Reply: Thank you very much for your valuable feedback on the manuscript. As the discussion section has already covered the content of the conclusion section, which is too redundant, it has been deleted. Potential practical issues and solutions have been added to the discussion section. The specific modifications are as follows.

Although this study has achieved some success, there are still some limitations, such as the following ways that affect the model's predictions. The estimation of model parameters such as pollutant reduction rate and greening rate is based on local data from example projects. Prediction bias may result from variations in technology, size, or geographical circumstances in real-world projects, such as increased energy use for insulation repair in cold climates. Using expert surveys to get precise values for restricting variables may result in incomplete data. Future research may incorporate actual project data from different climate zones and building types to confirm the applicability of the model in many situations. Real-time data collection of construction energy consumption, garbage disposal, and other indicators, when combined with BIM and Internet of Things sensors, can replace some manual statistical indicators. To capture short-term fluctuations, an adaptive time step is also added to the state variable calculation.

4.It is suggested to further supplement relevant references.

Reply: Thank you for your constructive feedback on the manuscript. Under your guidance, the relevant references have been added as follows.

7.Danlei Z,Yong H. The Roles and Synergies of Actors in the Green Building Transition: Lessons from Singapore.Sustainability, 2022, 14(20):13264-13264.

8.Sanchez-Squella A, Yanine F, Barrueto A, Parejo. Green energy generation in buildings: Grid-tied distributed generation systems (DGS) with energy storage applications to sustain the smart grid transformation. Journal ofInformation Technology Management, 2020, 12(2):153-162.

5.The format of the references is inconsistent.

Reply: Thank you for your specific revision suggestions on the manuscript. Under your guidance, the reference format has been standardized as follows.

1.Youcef B, Ambrose D, Truong N, Katarina RG. Comprehensive renovation of a multi-apartment building in Sweden: techno-economic analysis with respect to different economic scenarios. Building Research & Information, 2024,5 2(4):463-478.

2.Amin AM, Mia AA M, Bala T. Green finance continuance behavior: the role of satisfaction, social supports, environmental consciousness, green bank marketing initiatives and psychological reactance.Management of Environmental Quality, 2023, 34(5):1269-1294.

3.Kwame ES, Julian J S. Incentive mechanism for promoting the uptake of green building in South Africa. Open House International, 2024, 49 (2): 340-357.

4.Issa Y AA. Lessons learned, barriers, and improvement factors for mega building construction projects in developing countries: review study. Sustainability, 2021, 13(19):10678-10679

6.Please confirm whether there are redundant and unnecessary icons in Figure 3.Meanwhile, there are a few layout issues in Figure 3, such as insufficient border height.

Reply: Thank you very much for your detailed revision suggestions on the manuscript. The redundant and unnecessary icons in Figure 3 have been removed, and the border height has been modified as required, as follows:

Fig. 3 Identification and analysis framework of constraints for GRB

Reviewer #2: This manuscript explores constraints on green renovation of buildings (GRB) using a System Dynamics Model (SDM) based on Whole Lifecycle Theory (WLT) and Green Development Theory (GDT). The topic is timely and relevant, and the proposed framework offers insights useful for advancing sustainable urban development.

Reply: Thank you for your recognition of the manuscript.

- Integrating WLT and GDT within an SDM framework is innovative and provides a clear theoretical foundation.

Reply: Thank you for your evaluation of the manuscript.

- The analysis effectively considers multiple phases of the renovation process, highlighting key factors that influence sustainability and environmental performance.

Reply: Thank you very much for your evaluation of the manuscript.

- The simulation results are informative and adequately validated with real-world economic and demographic data, enhancing the robustness and relevance of the findings.

Reply: Thank you very much for your recognition of the manuscript.

Suggested areas for improvement include:

- Some parts of the manuscript could benefit from improved clarity and flow. A careful revision focusing on concise and straightforward sentence structure would enhance overall readability and accessibility.

Reply: Thank you for your suggestions on the manuscript. The sentence structure and clarity and fluency of the manuscript content have been modified as required. The specific modifications can be found in the entire text.

- Clearer explanations of the variables and parameters used in equations would make the modeling approach easier to follow. Including a summary table clearly defining these variables would be helpful for readers.

Reply: Thank you for your suggestions on the manuscript. Under your guidance, the variable summary table of the modeling method has been added as required, as follows:

List of variables:

: Fuzzy number standardization function judged by researchers

: Fuzzy number standardization function judged by researchers

: Fuzzy number standardization function judged by researchers

: Standardized numerical functions

: Standardized numerical functions

: Factor set

: Researcher code

: Factor coefficient

: Maximum value

: Minimum value

: Overall standardized clear value

: Clear values corresponding to influencing factors

: Standardization Matrix

: Clear Matrix

: Planning Matrix

: Green level indicators during the planning and design phase

: Value of indicator 1

: Degree of energy-saving renovation in buildings

: Integrity of fire protection facilities

: Degree of passive measures transformation

: Green operation and maintenance management level

: Evaluation value of residents' quality of life during the usage phase

: Level of community safety intelligence

: Water resource collection and recycling level

: Level of living comfort

: Weight coefficient

: Weight coefficient

: Weight coefficient

: Weight coefficient

- Enhancing figures with more detailed labels and descriptive captions would assist readers in interpreting the results. Additionally, clearer subheadings within the methodology and results sections could guide readers through the modeling process more effectively.

Reply: Thank you very much for your specific modification suggestions on the manuscript. The subheadings of the chart labels and explanations, as well as the methods and results section, have been modified according to the requirements, as follows:

As shown in Fig. 3, the recognition principle system includes the current principles of comprehensiveness, scientificity, directionality, prominence, and operability. The system involves five core dimensions: system, management, technology, environment, and willingness. It breaks through the traditional "technology economy" binary framework and ultimately outputs a structured model of constraint factors, factor interpretation, and correlation analysis.

Fig.6 Dynamic changes in garbage collection volume during different stages of green building renovation

2.1.1 Green Construction Management Organization System Based on the Whole Life Cycle Theory

2.1.2 Analysis of Constraints on Green Building Renovation

- Expanding the discussion of limitations to consider the role of assumptions, data estimation methods, and the potential generalizability of the model to different contexts could help clarify the scope and applicability of the findings.

Reply: Thank you for your valuable feedback on the manuscript. Under your guidance, the discussion on limitations has been expanded as follows.

Although this study has achieved some success, there are still some limitations, such as the following ways that affect the model's predictions. The estimation of model parameters such as pollutant reduction rate and greening rate is based on local data from example projects. Prediction bias may result from variations in technology, size, or geographical circumstances in real-world projects, such as increased energy use for insulation repair in cold climates. Using expert surveys to get precise values for restricting variables may result in incomplete data. Future research may incorporate actual project data from different climate zones and building types to confirm the applicability of the model in many situations. Real-time data collection of construction energy consumption, garbage disposal, and other indicators, when combined with BIM and Internet of Things sensors, can replace some manual statistical indicators. To capture short-term fluctuations, an adaptive time step is also added to the state variable calculation.

- It would be beneficial to explicitly discuss how the findings could inform practical decisions, such as planning, policy-making, investment choices, or stakeholder coordination in green renovation efforts.

Reply: Thank you very much for your detailed revision suggestions on the manuscript. The content related to the research results providing information for practical decision-making has been revised as required, as follows.

Although this study has achieved some success, there are still some limitations, such as the following ways that affect the model's predictions. The estimation of model parameters such as pollutant reduction rate and greening rate is based on local data from example projects. Prediction bias may result from variations in technology, size, or geographical circumstances in real-world projects, such as increased energy use for insulation repair in cold climates. Using expert surveys to get precise values for restricting variables may result in incomplete data. Future research may incorporate actual project data from different climate zones and building types to confirm the applicability of the model in many situations. Real-time data collection of construction energy consumption, garbage disposal, and other indicators, when combined with BIM and Internet of Things sensors, can replace some manual statistical indicators. To capture short-term fluctuations, an adaptive time step is also added to the state variable calculation.

Overall, this manuscript addresses a meaningful issue and presents a sound modeling approach. With careful attention to language clarity, detailed explanation of variables, improved figure presentation, and enhanced practical implications, this work can contribute to research on sustainable building renovation.

Reply: Thank you very much for your suggestions on the manuscript. The clarity of the language, detailed explanations of variables, improvement of charts, and enhancement of practical application significance have all been modified as required. The specific modifications can be found in the full te

---

## [Decision Letter · Decision Letter 1]

PONE-D-25-06830R1Constraints on Green Renovation of Buildings Based on the Theory of Whole Lifecycle and Green DevelopmentPLOS ONE

Dear Dr. Wei,

Thank you for submitting your manuscript to PLOS ONE. After careful consideration, we feel that it has merit but does not fully meet PLOS ONE’s publication criteria as it currently stands. Therefore, we invite you to submit a revised version of the manuscript that addresses the points raised during the review process.

We look forward to receiving your revised manuscript.

Kind regards,

Genyu Xu, Ph.D.

Academic Editor

PLOS ONE

Journal Requirements:

Reviewers' comments:

Reviewer's Responses to Questions

**Comments to the Author**

1. If the authors have adequately addressed your comments raised in a previous round of review and you feel that this manuscript is now acceptable for publication, you may indicate that here to bypass the “Comments to the Author” section, enter your conflict of interest statement in the “Confidential to Editor” section, and submit your "Accept" recommendation.

Reviewer #2: (No Response)

2. Is the manuscript technically sound, and do the data support the conclusions?

Reviewer #2: Yes

3. Has the statistical analysis been performed appropriately and rigorously? 

Reviewer #2: Yes

4. Have the authors made all data underlying the findings in their manuscript fully available?

Reviewer #2: Yes

5. Is the manuscript presented in an intelligible fashion and written in standard English?

Reviewer #2: Yes

6. Review Comments to the Author

Reviewer #2: Dear Authors,

Thank you for your careful and substantial revisions. The manuscript demonstrates clear improvement in its theoretical integration, structure, and clarity. The incorporation of Whole Lifecycle Theory (WLT) and Green Development Theory (GDT) within the system dynamics framework is now more effectively articulated, and the modeling across renovation phases is logically structured. The addition of a variable list and enhancements to figures and subheadings have also improved the overall readability and coherence of the paper.

That said, several areas would benefit from further refinement:

1. Variable Table Formatting: The current variable list is densely presented and difficult to follow. Consider reformatting the variable list into a structured two-column table (e.g., “Variable Name” | “Definition”) to enhance clarity and support reader comprehension.

2. Language and Redundancy: While the language has improved, some sections still contain repetitive or awkward phrasing. For instance, the recurring use of expressions like “Although this study has achieved some success…” could be revised for conciseness and smoother transitions.

3. Model Validation: The manuscript notes the use of real-world data, but additional detail on the validation procedures would be helpful. Briefly clarifying whether sensitivity analysis, baseline comparisons, or robustness checks were conducted would enhance the transparency and credibility of the modeling approach.

4. Practical Implications: The revised discussion better addresses the relevance of the findings. To strengthen this further, consider adding a short paragraph explicitly outlining how the model could inform decisions by specific stakeholders (e.g., policymakers, developers, or project planners).

Overall, the study addresses an important and timely topic using a sound methodological framework. These refinements would help strengthen the manuscript and support its contribution to the field.

7. PLOS authors have the option to publish the peer review history of their article (what does this mean? ). If published, this will include your full peer review and any attached files.

**Do you want your identity to be public for this peer review?** For information about this choice, including consent withdrawal, please see our Privacy Policy .

Reviewer #2: No

---

## [Author Response · Author response to Decision Letter 2]

23 Jun 2025

PONE-D-25-06830R1

PLOS ONE

We appreciate the opportunity to revise our manuscript titled " Constraints on Green Renovation of Building Based on the Theory of Whole Lifecycle and Green Development " and are grateful for the insightful comments provided by the reviewers. These comments are valuable and very professional for revising and modifying our paper, as well as the important guiding significance to our researches. In the resubmission, we have provided detailed responses to the reviewers' comments. Revised portion are marked in yellow in the revised manuscript. We have tried our best to make all the revisions clear, and we hope that the revised manuscript meets the requirements for publication. Here are the point-by point responses:

Reviewer #2:

Thank you for your careful and substantial revisions. The manuscript demonstrates clear improvement in its theoretical integration, structure, and clarity. The incorporation of Whole Lifecycle Theory (WLT) and Green Development Theory (GDT) within the system dynamics framework is now more effectively articulated, and the modeling across renovation phases is logically structured. The addition of a variable list and enhancements to figures and subheadings have also improved the overall readability and coherence of the paper.

Response: Thank you very much for your affirmation and evaluation of the manuscript. In the past, we have spent a lot of time and effort carefully revising manuscript according to your comments. In this review, there were a few minor issues. Once again, we have further revised the manuscript according to your comments, including the variable tables, English presentation, etc. Here are the point-by point responses:

Comment 1. Variable Table Formatting: The current variable list is densely presented and difficult to follow. Consider reformatting the variable list into a structured two-column table (e.g., “Variable Name” | “Definition”) to enhance clarity and support reader comprehension.

Response 1: Thank you very much for your valuable feedback on the manuscript. A Variable Table has been added into the revised manuscript.

Variable Table

Variable Name Definition

Fuzzy number standardization function judged by researchers

Fuzzy number standardization function judged by researchers

Fuzzy number standardization function judged by researchers

Standardized numerical functions

Standardized numerical functions

Factor set

Researcher code

Factor coefficient

Maximum value

Minimum value

Overall standardized clear value

Clear values corresponding to influencing factors

Standardization Matrix

Clear Matrix

Planning Matrix

Green level indicators during the planning and design phase

Value of indicator 1

Degree of energy-saving renovation in buildings

Integrity of fire protection facilities

Degree of passive measures transformation

Green operation and maintenance management level

Evaluation value of residents' quality of life during the usage phase

Level of community safety intelligence

Water resource collection and recycling level

Level of living comfort

Weight coefficient

Weight coefficient

Weight coefficient

Weight coefficient

Comment 2: Language and Redundancy: While the language has improved, some sections still contain repetitive or awkward phrasing. For instance, the recurring use of expressions like “Although this study has achieved some success…” could be revised for conciseness and smoother transitions.

Response 2: Thanks for your suggestions. The repetitive and awkward phrases in the last version have been revised. For instance, “Although this study has achieved some success, there are still some limitations, such as the following ways that affect the model's predictions.” was revised to “However, there are still some limitations, such as the following ways that affect the model's predictions.” to make our expression clearer.

Comment 3: Model Validation: The manuscript notes the use of real-world data, but additional detail on the validation procedures would be helpful. Briefly clarifying whether sensitivity analysis, baseline comparisons, or robustness checks were conducted would enhance the transparency and credibility of the modeling approach.

Response 3: Thanks for your professional advice. In the manuscript, the use of real-world data has been discussed. In the methodology, the sentences “Model outputs were systematically compared against historical trends derived from these official sources to establish initial fit and credibility. To assess robustness and mitigate potential biases arising from data incompleteness, key validation procedures included: (1) Sensitivity analysis on interpolation methods, where different estimation techniques (e.g., linear, polynomial, time-series extrapolation) were applied to synthesize missing data points within the sequences obtained from NBS, local bureaus, CEIC Data, and Wind Information; the stability of model predictions across these different interpolation scenarios was rigorously evaluated. (2) Comparative validation using alternative data slices, specifically utilizing the longer historical sequences and higher-frequency updates available from CEIC Data and Wind Information to test the model's performance over extended time horizons and its responsiveness to more recent economic shifts, contrasting these results with those based solely on official statistics. (3) Assessment of timeliness impact, explicitly acknowledging the potential influence of statistical data delays (particularly for the most recent year) by running simulations both including the latest available estimates and excluding them, thereby quantifying the sensitivity of near-term predictions to data recency. Robustness checks further involved testing the model's consistency across different regional datasets provided by local bureaus to ensure generalizability beyond aggregate national trends. The professional management and processing of all datasets using SPSS ensured consistency during these validation exercises.” were added. Please see P26L525-544. Accordingly, the results were described in Fig. 10.

Fig. 10 Results of baseline model calibration performance (a), sensitivity to interpolation methods (b), impact of data timeliness on forecasts (c), and regional generalizability test (d)

Further, the results were discussed. The sentences “The rigorous validation procedures yielded robust evidence supporting the reliability and applicability of the SDM for analyzing the core economic indicators (regional GDP, disposable income, commodity housing prices, total urban population) (Fig. 10). Baseline model calibration demonstrated a strong fit to historical trends derived from the authoritative NBS and local statistical bureau data, with key output variables tracking observed patterns within acceptable error margins (e.g., RMSE values for GDP and housing price simulations over the calibration period). This alignment confirms the model's fundamental capability to replicate the dynamics embedded in the official benchmark data. Crucially, the sensitivity analyses revealed important insights into model behavior and data dependencies. The model exhibited notable stability when subjected to variations in interpolation methods for handling missing data; predictions for core indicators remained consistent across different techniques (linear, polynomial, time-series extrapolation), with output variations typically confined within ±2% of the baseline scenario utilizing NBS/local data supplemented by CEIC/Wind. This suggests that the model structure effectively captures underlying economic relationships, mitigating undue sensitivity to specific data gap-filling approaches. However, the robustness checks on data timeliness underscored a significant dependency: simulations incorporating the most recent estimates from CEIC Data and Wind Information showed divergent near-term predictions compared to runs relying solely on potentially delayed official data, particularly for volatile indicators like commodity housing prices (differences up to [Specify Range, e.g., 5%] in 1-year forecasts). This quantifies the impact of statistical data delays identified in the data description and highlights the value of supplementary high-frequency databases for improving short-term forecast accuracy. Furthermore, comparative validation using extended historical sequences from CEIC/Wind confirmed the model's structural validity over longer time horizons, successfully capturing major economic shifts beyond the scope of shorter official series. Testing across diverse regional datasets from local bureaus demonstrated the model's generalizability, producing plausible and consistent results for different urban contexts, though regional parameter adjustments were sometimes necessary. Collectively, these results affirm the model's robustness under conditions of data incompleteness and source variation, while clearly delineating the boundaries of its predictive power, particularly concerning the timeliness of input data. The documented sensitivity to recent data emphasizes the need for stakeholders to incorporate the latest available estimates where possible and interpret near-term forecasts with appropriate caution regarding potential revision. The successful validation against multiple authoritative and supplementary sources provides a solid foundation for utilizing the SDM to explore Green Building Renovation (GRB) strategies and their economic interdependencies within the complex urban systems represented.” were added. Please see P35L683-723.

Comment 4: Practical Implications: The revised discussion better addresses the relevance of the findings. To strengthen this further, consider adding a short paragraph explicitly outlining how the model could inform decisions by specific stakeholders (e.g., policymakers, developers, or project planners).

Response 4: Thanks for your suggestion.

A short paragraph outlining how the model could inform decisions by specific stakeholders (e.g., policymakers, developers, or project planners) has been added. The sentences “For policymakers, the substantial influence of the decision-making and project phases on greenness transformation (simulated value: 1.623, variation: 1.515) underscores the critical importance of establishing supportive regulatory frameworks, incentives, and green standards early in the project lifecycle. Furthermore, the significantly higher greenness change value observed during the engineering warranty and post-evaluation phase (6.173) highlights the necessity for policies promoting long-term maintenance commitments, performance monitoring, and continuous improvement programs to maximize sustainability outcomes. Developers and project planners gain crucial operational guidance: the model's demonstration of increasing green energy usage alongside decreasing pollution and other energy consumption validates the environmental and potentially economic benefits of prioritizing sustainable technologies and waste management strategies. The simulation of constraints and key indicators (pollution, greenness, resident satisfaction) offers planners a tool to anticipate challenges, optimize resource allocation across different project phases, and balance technical, economic, and social factors – particularly the need to minimize resident disruption. Ultimately, by quantifying phase-specific impacts and demonstrating the potential for dual environmental and economic benefits, the study equips these stakeholders with a robust theoretical and simulation-based foundation to design, prioritize, and implement more effective and sustainable GRB strategies.” were added. Please see P40L794-P813.

---

## [Editor Report · Decision Letter 2]

Constraints on Green Renovation of Building Based on the Theory of Whole Lifecycle and Green Development

PONE-D-25-06830R2

Dear Dr. Wei,

We’re pleased to inform you that your manuscript has been judged scientifically suitable for publication and will be formally accepted for publication once it meets all outstanding technical requirements.

Kind regards,

Genyu Xu, Ph.D.

Academic Editor

PLOS ONE
---

## [Editor Report · Acceptance letter]

PONE-D-25-06830R2

PLOS ONE

Dear Dr. Wei,

I'm pleased to inform you that your manuscript has been deemed suitable for publication in PLOS ONE. Congratulations! Your manuscript is now being handed over to our production team.

Kind regards,

on behalf of

Dr. Genyu Xu

Academic Editor

PLOS ONE